# NEO — No-Optimization Test-Time Adaptation through Latent Re-Centering

**Alexander Murphy**[1][*]   **Michal Danilowski**[1]   **Soumyajit Chatterjee**[23]   **Abhirup Ghosh**[13]
[1]University of Birmingham   [2]Brave Software Research   [3]University of Cambridge

## Abstract

Test-Time Adaptation (TTA) methods are often computationally expensive, require a large amount of data for effective adaptation, or are brittle to hyperparameters. Based on a theoretical foundation of the geometry of the latent space, we are able to significantly improve the alignment between source and distribution-shifted samples by re-centering target data embeddings at the origin. This insight motivates `NEO` – a hyperparameter-free fully TTA method, that adds no significant compute compared to vanilla inference. `NEO` is able to improve the classification accuracy of ViT-Base on ImageNet-C from 55.6% to 59.2% after adapting on just one batch of 64 samples. When adapting on 512 samples `NEO` beats all 7 TTA methods we compare against on ImageNet-C, ImageNet-R and ImageNet-S and beats 6/7 on CIFAR-10-C, while using the least amount of compute. `NEO` performs well on model calibration metrics and additionally is able to adapt from 1 class to improve accuracy on 999 other classes in ImageNet-C. On Raspberry Pi and Jetson Orin Nano devices, `NEO` reduces inference time by 63% and memory usage by 9% compared to baselines. Our results based on 3 ViT architectures and 4 datasets show that `NEO` can be used efficiently and effectively for TTA.

## 1 Introduction

A central challenge in machine learning is maintaining performance under distribution shifts between training and deployment. For instance, an image classifier may excel on curated training data but degrade on real-world inputs with snow, fog, or motion blur. Test-Time Adaptation (TTA) methods (Li et al., 2018; Wang et al., 2024; Liang et al., 2020; Wang et al., 2021; Niu et al., 2023) address this by leveraging unlabeled test samples without requiring access to training data, making them particularly suited to the modern setting of large pre-trained models.

Existing TTA methods face several limitations, such as backpropagation-based updates that significantly increase memory consumption (Wang et al., 2022; Ma et al., 2025), inference latency (Niu et al., 2024; 2023), and sensitivity to hyperparameter choices (Wang et al., 2021). Others impose architectural assumptions (e.g., the presence of batch normalization layers) (Wang et al., 2021; Niu et al., 2023; Song et al., 2023) or require a large number of target samples to achieve stable adaptation (Iwasawa & Matsuo, 2021). Moreover, as adaptation and inference are performed continually on data arrival, TTA methods with high computational demands incur significant latency and memory overhead on both edge and server deployments.

We propose `NEO`, an optimization and hyperparameter-free fully (not using source data) TTA method with no significant additional latency or memory overhead. Moreover, `NEO` is more accurate and better calibrated than baseline TTA methods, which use up to several times the compute, as shown in Figure 2b. It is robust with the capability to adapt with just a single sample and with unbalanced classes for considerable accuracy improvements as shown in Figure 5.

We observe that due to the distribution shift at input, including data-independent stochastic shifts, the activations from the penultimate layer, *embeddings*, shift structurally. Our key idea is to approximate this shift by tracking the displacement of the global centroid of the embeddings and then re-centering test-time features accordingly. In practice, this means that given a corrupted sample $\tilde{x}$

---

[*]Correspondence to: `alexandermurphy784@gmail.com`

```
# Classifier head
self.classifier = nn.Linear(config.hidden_size, config.num_labels)

# NEO classifier head
self.classifier = NEO(config.hidden_size, config.num_labels)
```

Figure 1: Elegant adoption: `NEO` can be added by replacing the `nn.Linear` with our custom layer.

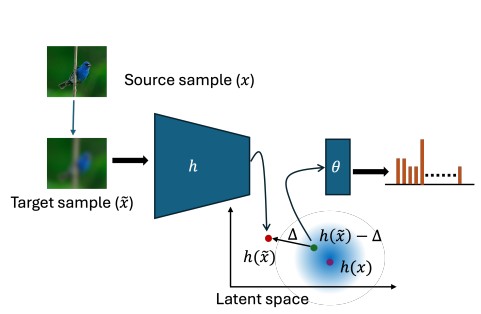

(a) High-level overview of `NEO`

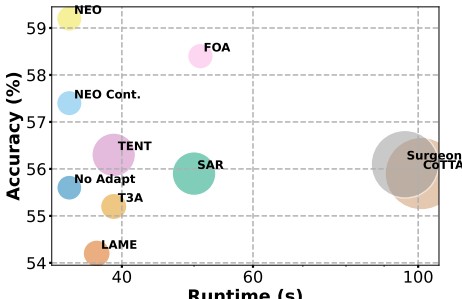

(b) `NEO` improves accuracy using little latency or memory

Figure 2: **(a)** Given a domain shifted sample, $\tilde{x}$, we encode it to $h(\tilde{x})$ and shift it using a single shared vector $\Delta$. The shifted representation is closer to the embedding of the corresponding clean sample (unknown), $h(x)$, resulting in more accurate predictions. **(b)** Runtime (x axis), accuracy (y axis), and memory usage (point radius) of TTA methods for ViT-Base on 15 corruption from ImageNet-C evaluated on 512 samples per corruption. `NEO` outperforms all methods in terms of runtime, accuracy, and memory.

with embedding $h(\tilde{x})$, `NEO` corrects it by subtracting a single shared vector $\Delta$, bringing the representation closer to the embedding of its clean counterpart $h(x)$ (although unknown) and thereby restoring accuracy – we illustrate the re-centering in Figure 2a. While estimating such a distribution shift is challenging in pre-trained models, often trained on unknown data sources, we leverage structural properties of neural collapse (Papyan et al., 2020) to develop a practical and principled method, without relying on source data.

`NEO` is elegant and simple to adopt, requiring only a single line change in a standard Vision Transformer implementation, by replacing `nn.Linear` with our custom `NEO` layer as shown in Figure 1. It incurs negligible computational and memory overhead as it stores only a single vector to correct the shift. Furthermore, with just a handful of samples, it consistently improves accuracy over state-of-the-art TTA methods that demand orders of magnitude more compute and memory. Finally, we also extend `NEO` with a continual variant, using just one easily interpretable hyperparameter, that maintains robustness under evolving test distributions, making it particularly suitable for real-world, resource-constrained deployments.

1. We propose `NEO`, a lightweight TTA method, that re-centers embeddings using a global centroid estimate. We evaluate `NEO` on 4 datasets and 3 different ViT architectures, showing consistent improvement in accuracy and model calibration. Moreover, `NEO` beats all 7 TTA methods that we compare against, when adapting on 512 samples from ImageNet-C.

2. We perform a thorough investigation on the effect that an input distribution shift has on the latent space, finding a significant shared shift across samples and classes. We connect this shift with neural collapse to provide a principled explanation for why global re-centering is sufficient for adaptation.

3. We show that `NEO` is both efficient and versatile: it can adapt with as little as a single sample or class, extend naturally to continual adaptation across evolving corruptions, and maintain low latency and memory usage on both edge devices and cloud servers. Combined with being hyperparameter-free, this makes `NEO` practical for diverse real-world deployment scenarios.

## 2 PROBLEM STATEMENT AND SETUP

Consider a pre-trained classification model $f : \mathbb{R}^m \to \mathbb{R}^C$, composed of an encoder $h : \mathbb{R}^m \to \mathbb{R}^d$ and a linear classification head $\theta : \mathbb{R}^d \to \mathbb{R}^C$, such that $f = \theta \circ h$. The model is trained on a source dataset $\mathcal{D} = (\boldsymbol{X}, \boldsymbol{Y})$, where $\boldsymbol{X} \in \mathbb{R}^{n \times m}$ contains $n$ labeled training samples. We aim to adapt $f$ to a domain-shifted target dataset $\tilde{\mathcal{D}} = (\tilde{\boldsymbol{X}}, \tilde{\boldsymbol{Y}})$, where $\tilde{\boldsymbol{X}} \in \mathbb{R}^{n' \times m}$ contains $n'$ target input samples, and $\tilde{\boldsymbol{Y}} \in \mathbb{R}^{n' \times C}$ contains the associated labels. We focus on a covariate shift that changes $\tilde{\boldsymbol{X}}$ and leaves the conditional distribution of the labels on input data unchanged.

A fully TTA algorithm $\mathcal{A}$ adapts the model $f$ by accessing just the target samples $\tilde{\boldsymbol{X}}$ without access to source data or target labels, such that $f_{adapt} = \mathcal{A}(f, \tilde{\boldsymbol{X}})$. The goal of $\mathcal{A}$ is to optimize an evaluation metric $\xi(f_{adapt}, \tilde{\boldsymbol{X}}, \tilde{\boldsymbol{Y}})$, such as accuracy. The evaluation metric may access the target labels, but $\mathcal{A}$ does not. We focus on two metrics: accuracy and expected calibration error (ECE) (Pakdaman Naeini et al., 2015). ECE quantifies the mismatch between predicted confidence and true accuracy which is important for evaluating model trustworthiness (see more in Appendix B.4).

TTA is useful in practical systems requiring adaptation to in-the-wild data distributions. For example, consider an on-car camera-based traffic signal recognizer that feeds into the car dashboard. Here, the application requires the model to be on-car for robustness against network failures. The model also needs to be adapted to remain accurate on images captured by the local camera. To meet real-time inference requirements and limited on-car compute resources, the adaptation needs to be time and resource-efficient, which forms the constraints we focus on in this paper.

## 3 RELATED WORKS AND BACKGROUND

A host of existing TTA methods minimize classification entropy through updating affine parameters of batch normalization (BN) layers, including TENT (Wang et al., 2021), SAR (Niu et al., 2023), and EATA (Niu et al., 2022). This line of work has been complemented by pseudo-label-based clustering (Liang et al., 2020), non-i.i.d. adaptation (Gong et al., 2022), and continual adaptation (Wang et al., 2022).

Despite their effectiveness, the above methods are computationally intensive as they rely on optimization through backpropagation (Danilowski et al., 2025). There are three primary approaches of efficient TTA methods: firstly, reducing the memory footprint by selecting important layers to adapt (Ma et al., 2025), secondly, using custom model architectures (Song et al., 2023; Hong et al., 2023; Jia et al., 2024), and finally, avoiding backpropagation altogether. The final approach is the most efficient, and there are two broad ways to achieve it: $i$) optimization through only forward passes using iterative

Table 1: Comparisons of selected baselines. FTTA: Fully TTA without using source data. HF: hyperparameter free, OF: optimization free, BI: Batch size independence, BF: backpropagation free, FP: # of forward propagations.

|  | FTTA | HF | OF | BI | BF | # FP |
|---|---|---|---|---|---|---|
| T3A | ✓ | ✗ | ✓ | ✗ | ✓ | 1 |
| SAR | ✓ | ✗ | ✗ | ✗ | ✗ | 2 |
| LAME | ✓ | ✓ | ✗ | ✗ | ✓ | 1 |
| TENT | ✓ | ✗ | ✗ | ✗ | ✗ | 1 |
| CoTTA | ✓ | ✗ | ✗ | ✗ | ✗ | 35 |
| FOA[1] | ✗ | ✗ | ✗ | ✗ | ✓ | 2 |
| Surgeon | ✓ | ✗ | ✗ | ✗ | ✗ | 3 |
| NEO-Cont. (ours) | ✓ | ✗ | ✓ | ✗ | ✓ | 1 |
| NEO (ours) | ✓ | ✓ | ✓ | ✓ | ✓ | 1 |

optimizers (Niu et al., 2024; Dong et al., 2025) and $ii$) adapt analytically without optimization. For example, adjusting BN statistics (Schneider et al., 2020; Nado et al., 2020; Su et al., 2024) or classifier adaptation via prototypes or outputs (Wang et al., 2024; Iwasawa & Matsuo, 2021; Boudiaf et al., 2022). NEO belongs to the latter, most efficient category, avoiding optimizers wholly.

An analytical solution to TTA is hard, due to the stochastic nature of the input noise and the complexity of modern neural architectures. Prior optimization-free methods require large batch sizes along with a large dataset to compute robust statistics, which is essential for their algorithms. Some

---

[1]FOA uses 32 samples to compute statistics from the source data.

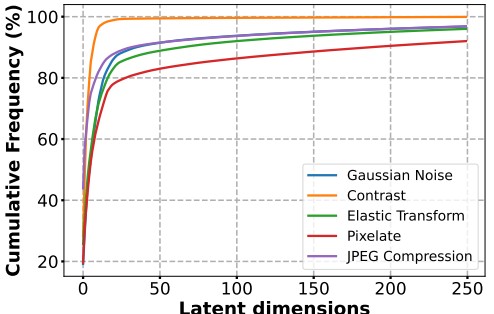

| Vector compared with $h(\boldsymbol{x})$ | Cos. | L2 Diff. |
|---|---|---|
| $h(\tilde{\boldsymbol{x}})$ | $-0.44$ | 4.33 |
| $h(\tilde{\boldsymbol{x}}) - \boldsymbol{\Delta}_G$ | 0.51 | 3.65 |
| $h(\tilde{\boldsymbol{x}}) - \boldsymbol{\Delta}_G - \boldsymbol{\Delta}_c$ | 0.64 | 5.47 |
| $h(\tilde{\boldsymbol{x}}) - \boldsymbol{\Delta}_G - \boldsymbol{\Delta}_c - \delta$ | 1.00 | 0.00 |
| $h(\tilde{\boldsymbol{x}}) - \widetilde{\boldsymbol{\mu}}_G$ | 0.49 | 3.64 |

(a) Few high magnitude dimensions form the shift

(b) Different shifts improve cosine similarity and L2 norm difference between corrupted and source data

Figure 3: **(a)** Cumulative frequency of highest magnitude dimension in $h(\boldsymbol{x}) - h(\tilde{\boldsymbol{x}})$ over 50000 samples (showing 250 out of 768 dimensions). A small number of dimensions account for the largest magnitude of the difference between source and corrupted embeddings. **(b)** Cosine similarities and difference of L2 norms between source embeddings and (adjusted) corrupted embeddings (i.e. first row contains average of $cos(h(\boldsymbol{x}), h(\tilde{\boldsymbol{x}}))$ and average of $|\,\|h(\boldsymbol{x})\| - \|h(\tilde{\boldsymbol{x}})\|\,|$). Embeddings are taken from ImageNet-C severity level 5 Gaussian Noise, on ViT-Base model (pre-trained on ImageNet). Values are averaged over 50000 samples.

existing TTA methods, use components which center embeddings at the center of source data, but rely on available source data to do so (Niu et al., 2024). We find a surprising structure in the change in the latent geometry, which we connect through the neural collapse phenomenon (Papyan et al., 2020), to make `NEO` fully independent of training data. Neural collapse emerges in the terminal phase of training and formalizes the geometric structure of the classifier weights and embeddings (see more in Appendix A.2). While this has been explored for domain generalization (Chen et al., 2024) and detecting out-of-distribution samples (Harun et al., 2025), as per our knowledge, this has not been investigated in the context of TTA methods.

## 4    CHARACTERIZING INPUT DATA DISTRIBUTION SHIFT

In this section, we observe that input data distribution shifts introduce a structural change in the latent space.

### 4.1    LATENT SHIFT AFFECTS A FEW DIMENSIONS THE MOST

We find that the most significant change between the source $h(\boldsymbol{x})$ and corrupted embeddings, $h(\tilde{\boldsymbol{x}})$, occurs in just a few embedding dimensions. We calculate the dimension with the largest change in magnitude for each sample when comparing source and corrupted data. In Figure 3 (left), we show that for corruptions such as *contrast*, less than 20 out of 768 dimensions have the largest difference for 95% of samples. For all corruptions, 80% of data has less than 50 dimensions as their highest magnitude change, signifying the possible existence of a globally shared shift across samples and classes. Motivated by this we formalize the structural change in latent space and create the foundation for `NEO`.

### 4.2    CHARACTERIZING PER-SAMPLE LATENT SHIFT VECTORS

Let $\boldsymbol{X}_c \subseteq \boldsymbol{X}$ be the collection of samples from class $c$. The (global) centroid of the latent representations for samples from $\boldsymbol{X}$ is $\boldsymbol{\mu}_G$ and the class-wise centroid of the samples from class $c$, $\boldsymbol{X}_c$ is $\boldsymbol{\mu}_c$. Let $\tilde{\boldsymbol{\mu}}_G$ and $\tilde{\boldsymbol{\mu}}_c$ denote the same quantities for corrupt samples, $\tilde{\boldsymbol{X}}$ and $\tilde{\boldsymbol{X}}_c$.

Let us define the global centroid shift as $\boldsymbol{\Delta}_G = \tilde{\boldsymbol{\mu}}_G - \boldsymbol{\mu}_G$ and class centroid shift as $\boldsymbol{\Delta}_c = \tilde{\boldsymbol{\mu}}_c - \boldsymbol{\mu}_c - \boldsymbol{\Delta}_G$, while accounting for the global centroid shift. Using the global and class centroid shifts, we can define $h(\tilde{\boldsymbol{x}}) = h(\boldsymbol{x}) + \boldsymbol{\Delta}_G + \boldsymbol{\Delta}_c + \delta$, where $\delta$ is a sample-specific residual component and $\boldsymbol{x}$ is from class $c$.

In Figure 3a we show the cosine similarity between source data and corrupted data, that is aligned globally, class-wise and sample-wise. The largest increase in cosine similarity is caused by the global alignment, signifying the importance of globally aligning the corrupted latent space. Class-wise alignment increases cosine-similarity again, but only marginally when compared to the increase caused by global alignment. We also show the difference in norms between source and aligned corrupted embeddings. We can see that global alignment greatly reduces the difference in norms. Surprisingly, class-wise alignment increases the difference in norms significantly. This shows that even though cosine similarity may be increased by class-wise alignment, it does not guarantee that norms are similar to those of source embeddings.

Note that none of the alignments we tested above are computable in the fully TTA setting, as no source data or target labels are available. Thus, we also show the cosine similarity and difference of norms for a computable alignment in the last row, which centers corrupted data at the origin. The centering increases cosine-similarity nearly as much as the global shift does while delivering the smallest difference in norms to the source embeddings — this centering is the foundation for NEO.

### 4.3 ALIGNMENT AND ACCURACY

There is a strong link between linear classification, the cosine similarity between the classifier weights and the embeddings, and the L2 norm of the embeddings. To further analyze this connection, we draw on the theory of neural collapse (Papyan et al., 2020; Súkeník et al., 2023).

**Proposition 4.1.** *Consider a network $f$ with a linear classifier. Assume the model is trained to neural collapse with cross-entropy loss, weight regularization and uniformly distributed classes. Then given $\boldsymbol{w}_c$, the classifier weight vector corresponding to class $c$, we have*

$$y = \arg\max_c \boldsymbol{w}_c h(\boldsymbol{x}) + b_c \iff y = \arg\max_c \|\boldsymbol{w}_c\|\|h(\boldsymbol{x})\|cos(\boldsymbol{w}_c, h(\boldsymbol{x})).$$

Proposition 4.1 (proof in Appendix A.3) shows that under neural collapse assumptions with cross-entropy loss, the assigned class is solely determined by the cosine similarity of embeddings and classifier weights. If we have a large cosine similarity between $h(\boldsymbol{x})$ and $h(\tilde{\boldsymbol{x}})$, then the similarity between the classifier weights and $h(\tilde{\boldsymbol{x}})$ will be similar to that of the classifier weights and $h(\boldsymbol{x})$. Note that $\|h(\boldsymbol{x})\|$ does not influence the assigned class, as it is a constant factor for all classes, but does influence the logit distribution. Thus the difference in norms between $h(\boldsymbol{x})$ and $h(\tilde{\boldsymbol{x}})$ will influence the confidence of predictions.

### 4.4 CENTERING AT THE ORIGIN IS GLOBAL ALIGNMENT

In Figure 3b we show that centering the corrupted embeddings at the origin, significantly improves cosine similarity and difference of norms, compared with source embedding. This is not a given though; a neural network's embeddings could be centered at any point that is not the origin.

**Proposition 4.2.** *Consider a network $f$ exhibiting neural collapse and trained with cross-entropy loss and regularization. Under the assumption of the unconstrained features model (Mixon et al., 2022) (treating $h(\boldsymbol{x})$ as a freely optimizable variable) and balanced classes, we have $\Delta_G = \tilde{\boldsymbol{\mu}}_G - \boldsymbol{\mu}_G = \tilde{\boldsymbol{\mu}}_G$.*[2]

*Proof.* We define $\Delta_G = \tilde{\boldsymbol{\mu}}_G - \boldsymbol{\mu}_G$, as the difference between the mean embeddings of clean and corrupted data. It's been shown that under cross-entropy loss $\boldsymbol{\mu}_G = \boldsymbol{0}_d$ (Hong & Ling, 2024; Zhu et al., 2021). From this follows that $\Delta_G = \tilde{\boldsymbol{\mu}}_G - \boldsymbol{\mu}_G = \tilde{\boldsymbol{\mu}}_G$. □

Proposition 4.2 shows that under neural collapse assumptions, the global alignment is equivalent to centering at the origin. This underpins our empirical findings shown in Figure 3b, where we show that shifting $h(\tilde{\boldsymbol{x}})$ by $\Delta_G$ and by $\tilde{\boldsymbol{\mu}}_G$ result in nearly identical cosine similarities and L2 differences when comparing with $h(\boldsymbol{x})$. Using the theoretical foundation we developed, we now propose NEO.

---

[2]We also provide the results for MSE loss in Appendix A.1.

## 4.5 NEO

Having established that $h(\tilde{\boldsymbol{x}}) - \tilde{\boldsymbol{\mu}}_G$ has desirable properties when fed into the linear classifier, we present NEO, which works on this exact principle. The global mean of corrupted embeddings is updated with each new batch of data, weighting all samples it sees equally, under the assumption that all test-time samples come from the same distribution. This global mean of corrupted embedding is then used to center the embeddings at the origin.

For line 4 in the pseudo-code, the Avg operator is used to turn the matrix of embedding vectors returned by the encoder function $h$ into a single embedding vector averaged across $b$ samples in the batch. The feature extractor and classifier are slightly modified from the previous sections, as they are able to process batches of data. It is easy to see that no significant computational requirements are added by NEO. The only operations involving vectors or matrices are averaging, addition, scalar multiplication and a single expansion of a vector into a matrix.

---

**Algorithm 1** NEO

**Require:** Dataset $\tilde{\boldsymbol{X}}$, feature extractor $h$, classifier $\theta$
1: $\tilde{\boldsymbol{\mu}}_G \leftarrow \boldsymbol{0}_d, i \leftarrow 0$
2: **for** each batch $\boldsymbol{B} \in \mathbb{R}^{b \times m}$ in $\tilde{\boldsymbol{X}}$ **do**
3: $\quad i \leftarrow i + 1$
4: $\quad \tilde{\boldsymbol{\mu}}_G \leftarrow (i-1)/i \cdot \tilde{\boldsymbol{\mu}}_G + 1/i \cdot \mathrm{Avg}(h(\boldsymbol{B}))$
5: $\quad \boldsymbol{y} = \theta(h(\boldsymbol{B}) - \tilde{\boldsymbol{\mu}}_G \boldsymbol{1}_b^T)$
6: **end for**

---

NEO is robust in various ways. Various existing TTA methods try to approximate shifted class centers (Iwasawa & Matsuo, 2021), which results in few samples per class-center, and unreliable estimates. NEO approximates just one global shift and is able to use every sample to estimate this shift, making it a very robust estimate. Moreover, methods that rely on pseudo-labeling (Liang et al., 2020) can cause a reduction in accuracy when pseudo-labels are inaccurate (Wang et al., 2022). NEO does not rely on labels, thus adding another layer of robustness. Furthermore, the weights of $h$ stay unchanged, preventing catastrophic forgetting, that may occur in other methods (Niu et al., 2022). Lastly, NEO does not depend on the size of batches, just the amount of data it sees throughout adaptation, taking any samples into account equally and independently, making it reliable in settings where large batches are unavailable.

Since NEO weights all samples equally, it may be unsuitable for scenarios where the distribution shift changes. Thus we propose NEO-Continual, which uses the same equation to realign the test-time features to the source features, but uses an exponential moving average, controlled by a hyperparameter $\alpha$, to keep track of the test-time feature simplex mean, making it suitable for continual adaptation problems. We simply replace the update rule from NEO with:

$$\widetilde{\boldsymbol{\mu}}_G \leftarrow (1 - \alpha) \cdot \widetilde{\boldsymbol{\mu}}_G + \alpha \cdot \mathrm{Avg}(h(\boldsymbol{B})).$$

## 5 EXPERIMENTS

In this section, we first introduce baseline TTA methods, datasets, models and devices, we use in our experiments. We then perform multiple experiments to show the effectiveness of our proposed method and also why and how it works. Even though our motivation has made theoretical assumptions such as the existence of neural collapse, we do not require this for our experiments and use regular, publicly available models and datasets.

**Baseline TTA Methods.** We compare our method with popular adaptation methods: T3A (Iwasawa & Matsuo, 2021), SAR (Niu et al., 2023), LAME (Boudiaf et al., 2022), TENT (Wang et al., 2021), CoTTA (Wang et al., 2022), FOA (Niu et al., 2024) and Surgeon (Ma et al., 2025). The hyperparameters used are taken directly from their original paper, or slightly adjusted when default hyperparameters cause catastrophic forgetting on our experiment setup. We use a batch size of 64 for all experiments, except those investigating adaptation sample size, class distribution and resource utilization.

**Datasets.** We evaluate on ImageNet-C (50 samples $\times$ 1000 classes $\times$ 15 corruption types) (Hendrycks & Dietterich, 2019), CIFAR-10-C (1000 samples $\times$ 10 classes $\times$ 15 corruption types)

Table 2: Accuracy (%) with 95% confidence intervals across different corruption types and adaptation methods with ViT-Base on ImageNet-C. Accuracy is calculated on the 512 samples used to adapt. The highest accuracy per corruption type is in bold, and the second-highest is underlined.

| Corruption | No Adapt | T3A | SAR | LAME | TENT | CoTTA | FOA | Surgeon | NEO |
|---|---|---|---|---|---|---|---|---|---|
| *Noise* | | | | | | | | | |
| Gaussian | 57.0 (0.5) | 56.7 (1.7) | 57.0 (1.7) | 56.5 (1.7) | 57.2 (1.6) | 57.0 (1.6) | 57.2 (0.6) | **58.7 (0.6)** | 57.7 (0.5) |
| Shot | 56.9 (0.5) | 57.0 (1.0) | 57.3 (0.9) | 56.8 (1.0) | 57.5 (1.0) | 57.1 (1.1) | 58.6 (0.4) | **58.8 (0.5)** | 57.6 (0.5) |
| Impulse | 57.4 (0.4) | 57.0 (1.0) | 57.5 (1.1) | 56.7 (0.9) | 57.6 (1.0) | 57.7 (1.1) | 57.7 (0.4) | **58.9 (0.6)** | 58.1 (0.4) |
| *Blur* | | | | | | | | | |
| Defocus | 46.9 (0.5) | 47.5 (1.2) | 47.5 (1.2) | 47.1 (1.1) | 48.0 (1.2) | 48.2 (1.2) | 49.2 (0.4) | 49.1 (0.8) | **49.8 (0.5)** |
| Glass | 35.3 (0.5) | 35.9 (0.9) | 36.3 (0.9) | 34.9 (1.0) | 36.8 (0.8) | 36.0 (1.0) | 36.8 (0.5) | 36.8 (0.7) | **37.9 (0.4)** |
| Motion | 53.3 (0.4) | 53.1 (1.1) | 53.5 (1.1) | 52.9 (1.0) | 54.0 (1.0) | 53.3 (1.1) | 54.4 (0.4) | 54.8 (0.6) | **55.0 (0.4)** |
| Zoom | 44.8 (0.5) | 45.3 (1.4) | 46.3 (1.5) | 44.7 (1.3) | 46.4 (1.4) | 45.0 (1.4) | 47.1 (0.5) | 45.7 (0.6) | **47.5 (0.5)** |
| *Weather* | | | | | | | | | |
| Snow | 62.2 (0.5) | 62.7 (1.7) | 62.6 (1.5) | 58.5 (1.6) | 63.0 (1.6) | 63.2 (1.4) | 64.3 (0.4) | 62.2 (0.7) | **64.6 (0.5)** |
| Frost | 62.6 (0.5) | 63.3 (1.5) | 63.3 (1.5) | 62.2 (1.5) | 63.3 (1.4) | 63.0 (1.4) | 63.9 (0.4) | 61.7 (0.6) | **65.0 (0.5)** |
| Fog | 65.8 (0.4) | 62.9 (1.2) | 65.4 (1.1) | 62.0 (1.0) | 62.4 (1.3) | 64.8 (1.0) | 70.7 (0.4) | 63.0 (0.6) | **71.2 (0.4)** |
| Brightness | 77.9 (0.4) | 78.1 (1.1) | 78.0 (1.1) | 78.1 (1.2) | 78.2 (1.2) | 77.9 (0.8) | 78.2 (0.4) | 78.1 (0.5) | **78.3 (0.4)** |
| *Digital* | | | | | | | | | |
| Contrast | 32.6 (0.4) | 27.5 (1.2) | 34.0 (1.3) | 24.9 (1.4) | 36.9 (1.0) | 33.2 (1.0) | 54.5 (0.5) | 31.7 (1.5) | **58.2 (0.4)** |
| Elastic | 45.8 (0.4) | 45.8 (0.9) | 45.7 (0.9) | 44.4 (0.7) | 46.7 (0.8) | 46.3 (1.2) | 49.6 (0.4) | 46.2 (0.7) | **49.8 (0.4)** |
| Pixelate | 67.5 (0.4) | 67.4 (1.2) | 67.5 (1.1) | 67.1 (1.1) | 68.0 (1.1) | 67.8 (1.1) | 67.2 (0.4) | 67.6 (0.7) | **68.2 (0.4)** |
| JPEG | 67.9 (0.4) | 67.3 (1.5) | 67.3 (1.5) | 66.9 (1.4) | 67.6 (1.5) | 67.8 (1.7) | 68.6 (0.5) | 68.9 (0.7) | **69.1 (0.4)** |
| **ImageNet-C** | 55.6 (0.4) | 55.2 (1.3) | 55.9 (1.2) | 54.2 (1.2) | 56.3 (1.2) | 55.9 (1.2) | 58.4 (0.4) | 56.1 (0.7) | **59.2 (0.4)** |
| **CIFAR-10-C** | 80.4 (2.8) | 80.1 (2.6) | 80.6 (2.8) | 79.8 (2.9) | 81.3 (3.2) | 80.6 (2.0) | 80.9 (2.8) | **82.7 (1.7)** | 82.4 (2.2) |
| **ImageNet-R** | 59.2 (1.1) | 58.7 (1.2) | 59.3 (1.1) | 58.5 (1.1) | 59.4 (1.1) | 59.2 (1.2) | 60.2 (1.4) | 60.2 (1.6) | **60.3 (1.0)** |
| **ImageNet-S** | 45.4 (1.4) | 45.5 (1.4) | 45.5 (1.4) | 45.0 (1.3) | 45.7 (1.4) | 45.2 (1.6) | 46.3 (1.7) | 47.0 (1.7) | **47.2 (1.4)** |

(Hendrycks & Dietterich, 2019), ImageNet-Rendition (30,000 samples, 200 classes) (Hendrycks et al., 2021) and ImageNet-Sketch (50 samples × 1000 classes) (Wang et al., 2019).

**Models.** In our experiments we use three different sizes of Vision Transformer (ViT) (Dosovitskiy et al., 2021): ViT-S, ViT-Base and ViT-L, which have 22, 86, and 307 million parameters and embedding dimensions of 384, 768 and 1024 respectively. We use versions finetuned on ImageNet (Deng et al., 2009) or CIFAR-10 (Krizhevsky & Hinton, 2009).

**Metrics.** We evaluate accuracy in two ways. Firstly, the accuracy achieved on samples used during the adaptation process. Secondly, the accuracy achieved on data not used for adaptation, after finishing the adaptation process. The type of accuracy used is stated in experiment descriptions. We use expected calibration error (ECE) (Pakdaman Naeini et al., 2015) to evaluate the trustworthiness of the model (i.e. does 90% confidence translate to 90% accuracy), where a lower ECE is better.

**Devices.** In our experiments, we evaluate on a Raspberry Pi 4B (8GB) and an NVIDIA Jetson Orin Nano (8GB), comparing TTA methods in terms of memory consumption and execution time. We also evaluate on a cloud server with an AMD Instinct MI300X (192GB VRAM).

For more details on implementation, including code, see Appendix B.

## 5.1 RESULTS

**NEO improves accuracy across corruption types.** Table 2 shows accuracy results over 15 corruption types in ImageNet-C for NEO compared to 7 baselines. Under 12 corruptions NEO has the highest accuracy and the second highest in the remaining 3, only beaten by Surgeon, which uses considerably more runtime and memory. On average NEO increases performance by 3.6% and in the contrast corruption NEO can nearly double accuracy compared to using no adaptation. It is also notable that NEO does not reduce accuracy under any corruption type. It has been shown that TTA is often sensitive to hyperparameters and can suffer from catastrophic forgetting, thus delivering near-zero accuracy (Niu et al., 2023) (Kojima et al., 2023). NEO is hyperparameter-free and thus extremely robust compared to other TTA methods.

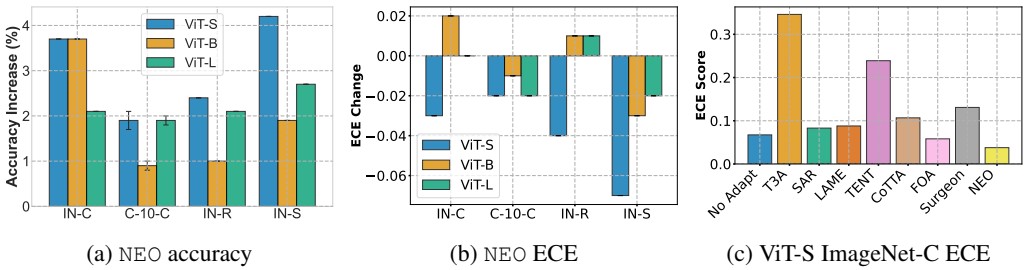

Figure 4: **(a)** Accuracy increase (%) and **(b)** ECE change compared to no-adaptation for ViT-S, ViT-B and ViT-L on ImageNet-C, CIFAR-10-C, ImageNet-Sketch and ImageNet-Rendition. Accuracy is taken for the whole dataset and no confidence intervals signify a 95% confidence interval of less than 0.05 for accuracy and less than 0.005 for ECE. **(c)** ECE scores for ViT-S on ImageNet-C averaged over the whole dataset, 15 corruptions and multiple runs.

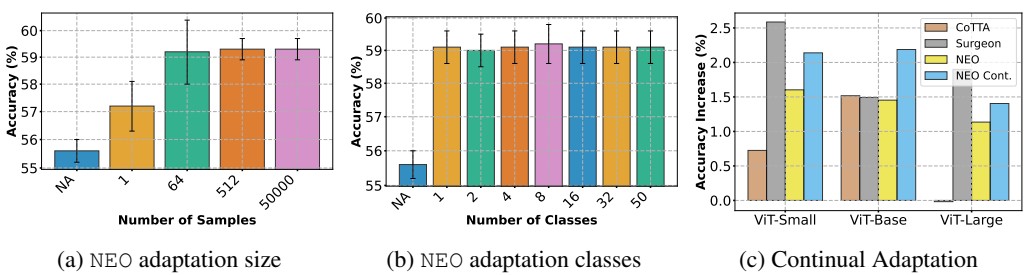

Figure 5: **(a)** Accuracy (%) for ViT-B on ImageNet-C under varying number of samples to adapt with. **(b)** Accuracy (%) for ViT-B on ImageNet-C under varying number of classes to adapt with (50 samples used to adapt in total). Accuracy is calculated on samples not used for adaptation except for 50,000 samples. **(c)** Accuracy increase (%) for continual adaptation, adapting on 15 randomly ordered corruptions from ImageNet-C with 512 samples from each.

**NEO improves accuracy and ECE across models and datasets.** In Figure 4 we show that NEO consistently improves accuracy across ImageNet-C, CIFAR-10-C, ImageNet-Sketch and ImageNet-Rendition, with an over 4% accuracy increase on Sketch. Across all ViT sizes NEO improves accuracy, with a trend of ViT-B consistently gaining less accuracy than ViT-S and ViT-L. When comparing with no adaptation, there is an improvement or match in ECE for 9/12 settings we evaluated. On ImageNet-C with ViT-S NEO achieves lower ECE than all compared TTA methods. This shows that NEO can be used to not only efficiently improve model performance, but also trustworthiness, by producing well-calibrated predictions.

**NEO can adapt with just 1 sample or class.** In some settings, samples to adapt with may be very limited, and not all classes may appear during adaptation. In Figure 5a, we show that adapting with just 1 sample is enough to improve accuracy by 1.5% on ImageNet-C. Adapting with just 1 batch (64 samples) is so effective that further samples improve accuracy only marginally. Figure 5b shows that adapting with data from just 1 class improves accuracy by over 3% on the remaining 999 classes in ImageNet-C. Adapting with 50 classes does not improve accuracy noticeably. Being able to adapt with few samples and classes shows that NEO is extremely robust and efficient.

**NEO is effective for continual adaptation.** We show the accuracy of the continual version of NEO in Figure 5c, where we present average accuracies while adapting on randomly ordered sequences of 15 corruptions from ImageNet-C. We use 512 samples from each corruption and the model is not informed when a corruption type is changed. NEO-Continual is more accurate than CoTTA and is only outperformed by Surgeon. Both of these methods use far more resources than NEO-Continual.

**Understanding $\tilde{\mu}_G$ across corruptions.** To understand the behavior of $\tilde{\mu}_G$ across corruption types, we $i$) adapted using a single domain and tested on the remaining 14 in ImageNet-C as shown

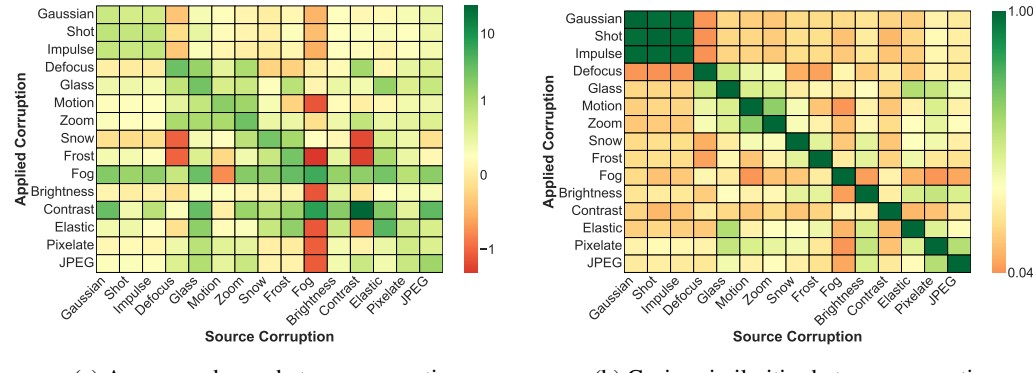

(a) Accuracy change between corruptions   (b) Cosine similarities between corruptions

Figure 6: **(a)** Accuracy change using $\tilde{\boldsymbol{\mu}}_G$ calculated from "Source Corruption" and adapting to samples from "Applied Corruption" **(b)** Cosine similarity between $\tilde{\boldsymbol{\mu}}_G$ calculated from "Source Corruption" and "Applied Corruption".

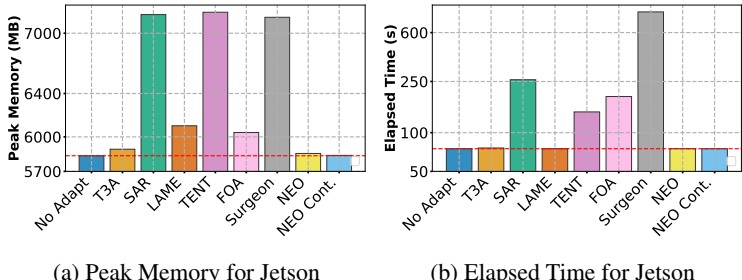

(a) Peak Memory for Jetson   (b) Elapsed Time for Jetson

Figure 7: Peak memory and elapsed time for adapting on Vit-Base on ImageNet-C (1000 samples - Gaussian Noise) with Nvidia Jetson Orin Nano.

in Figure 6a, and $ii$) measured cosine similarity of $\tilde{\boldsymbol{\mu}}_G$ between domains shown in Figure 6b. We discover that $\tilde{\boldsymbol{\mu}}_G$ has high accuracies and cosine similarities between certain corruptions such as noises or blurs. This has practical relevance as it can save resources needed to adapt to every domain observed.

**NEO is as efficient as no adaptation.** While Figure 2b shows that NEO adds no additional compute in a larger cloud environment, in Figure 7 we show efficiency results when running the methods on an edge device, Nvidia Jetson Orin Nano. NEO is the only method that does not increase the time used for inference, as well as the peak memory usage. Results on Raspberry Pi follow similar patterns and are in Appendix C.4.

## 6   CONCLUSIONS

In this paper, we present a novel yet simple TTA algorithm, NEO grounded on our insights on the geometry of latent space and the theory of neural collapse. It is extremely efficient on both server and edge devices and is robust to scarcity and bias in the adaptation dataset. Our simple implementation will help practitioners to adopt this method. We believe that our paper makes important contributions in two directions: the understanding of how embedding space is structured for neural networks and efficient test-time adaptation. We believe our work will trigger further development in both avenues.

**Limitations.** While NEO by design is not restricted to any specific model architecture, in line with the current technology trend, we evaluate on vision transformer architectures, and leave other architecture choices for the future. Our insights are inherently limited to understanding the activations from the penultimate layer and leave the investigations for other layers for future scope.

## 7 ACKNOWLEDGMENTS

Abhirup Ghosh and Alex Murphy acknowledge Royal Society Research Funding RGS\R2\242238 for supporting this research. We acknowledge Dr. Wendy Yanez Pazmino, for supporting us with edge device environments.

## 8 ETHICS STATEMENT

While NEO overall is an effective and efficient TTA method, there are ethical caveats, such as TTA being applied in settings that are illegal or immoral. Improved TTA performance could be misused to do harm in such situations. Of course, there are a range of positive impacts as well, such as reduced energy consumption and increased performance for in applications such as health and safety. Our code is publicly available to allow others to use NEO in the safest and most transparent way possible.

## 9 REPRODUCIBILITY STATEMENT

To aid with reproduction and transparency, we make our code publicly available here. We provide more details on reproduction, such as hyperparameters and evaluation metrics, in Appendix B.

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

# A  MORE THEORETICAL RESULTS

Here we present `NEO` for MSE, an explanation of neural collapse and the proof for Proposition 4.1.

## A.1  NEO FOR MSE

For models trained under MSE loss instead of cross-entropy loss, we need to adjust NEO.

**Proposition A.1.** *Consider a network $f$ exhibiting neural collapse and trained with MSE loss. Then, $\Delta_G = \tilde{\mu}$ and the bias of the classifier $\boldsymbol{b} = \frac{1}{C}\boldsymbol{1}_C$. Under the assumption of the unconstrained features model (Mixon et al., 2022) (treating $h(\boldsymbol{x})$ as a freely optimizable variable), we have*

$$\boldsymbol{W}(h(\tilde{\boldsymbol{x}}) - \tilde{\boldsymbol{\mu}}_G) + \frac{1}{C}\boldsymbol{1}_C = \boldsymbol{W}h(\boldsymbol{x}) + \boldsymbol{b}$$

.

*Proof.* Under the assumption of neural collapse, Papyan et al. (2020) have proven, using a result from Webb & Lowe (1990), that the ideal weights and bias of the classifier under mean square error loss and balanced classes are the following:

$$\boldsymbol{W} = \alpha\boldsymbol{M}^T,$$

$$\boldsymbol{b} = \frac{1}{C}\boldsymbol{1}_C - \alpha\boldsymbol{M}^T\boldsymbol{\mu}_G.$$

Thus, for the MSE case we subtract the shifted simplex mean and set the bias to $(1/C)\boldsymbol{1}_C$:

$$\boldsymbol{W}(h(\tilde{\boldsymbol{x}}) - \tilde{\boldsymbol{\mu}}_G) + \frac{1}{C}\boldsymbol{1}_C = \alpha\boldsymbol{M}^T(h(\boldsymbol{x}) + \tilde{\boldsymbol{\mu}}_G - \boldsymbol{\mu}_G - \tilde{\boldsymbol{\mu}}_G) + \frac{1}{C}\boldsymbol{1}_C = \boldsymbol{W}h(\boldsymbol{x}) + \boldsymbol{b}.$$

$\square$

The pseudo-code for NEO under MSE loss:

---

**Algorithm 2** NEO-MSE

---

**Require:** Dataset $S$, feature extractor $h$, classifier weights $\boldsymbol{W}$
1:   $\tilde{\boldsymbol{\mu}}_G \leftarrow \boldsymbol{0}_d, i \leftarrow 0$
2:   **for** each batch $\boldsymbol{B} \in \mathbb{R}^{b \times m}$ in $S$ **do**
3:      $i \leftarrow i + 1$
4:      $\tilde{\boldsymbol{\mu}}_G \leftarrow (i-1)/i \cdot \tilde{\boldsymbol{\mu}}_G + 1/i \cdot \text{mean}(h(\boldsymbol{B}))$
5:      $\boldsymbol{y} = \boldsymbol{W}(h(\boldsymbol{B}) - \tilde{\boldsymbol{\mu}}_G\boldsymbol{1}_b^T) + \frac{1}{C}\boldsymbol{1}_C\boldsymbol{1}_b^T$
6:   **end for**

---

## A.2  NEURAL COLLAPSE

Recent work analyzing the behavior of the last layer in neural networks, has discovered a phenomenon known as neural collapse (Papyan et al., 2020). It occurs when continuing to train a neural network after it has achieved near zero loss, also known as the terminal phase of training. It's a property of the last layer of the feature extractor, but recent work has also been expanding neural collapse to more layers in a neural network (Súkeník et al., 2023).

In addition to $\boldsymbol{\mu}_G$ and $\boldsymbol{\mu}_c$ which we defined in the problem statement, we also define the within-class covariance to be $\boldsymbol{\Sigma}_W = \text{Avg}_{i,c}\{(\boldsymbol{h}_{i,c} - \boldsymbol{\mu}_c)(\boldsymbol{h}_{i,c} - \boldsymbol{\mu}_c)^T\}$, where $\boldsymbol{h}_{i,c}$ is the embedding of sample $\boldsymbol{x}^{(i)}$ from class $c$. Papyan et al. (2020) identify the following four properties of neural collapse:

- **(NC1) Variability collapse:** The variation of features within the same class goes to near zero.
$$\boldsymbol{\Sigma}_W \to 0$$

- **(NC2) Convergence to simplex equiangular tight frame (ETF):** The class means of the features form the vertices of an ETF simplex, with equal length and angles between them. Essentially, all class means are an equal distance and at an equal angle from each other.

$$\big| \, \|\boldsymbol{\mu}_c - \boldsymbol{\mu}_G\|_2 - \|\boldsymbol{\mu}_{c'} - \boldsymbol{\mu}_G\|_2 \, \big| \to 0 \quad \forall c, c'$$

$$\langle \tilde{\mu}_c, \tilde{\mu}_{c'} \rangle \to \frac{C}{C-1} \delta_{c,c'} - \frac{1}{C-1} \quad \forall c, c'$$

- **(NC3) Convergence to self-duality:** The class means and classifier weights converge to each other upon rescaling.

$$\left\| \frac{\boldsymbol{W}^T}{\|\boldsymbol{W}\|_F} - \frac{\boldsymbol{M}^T}{\|\boldsymbol{M}\|_F} \right\|_F \to 0$$

- **(NC4) Simplification to nearest class-center:** At inference, the classifier solely decides which class to predict, by taking the class mean with the lowest euclidean distance.

$$\arg\max_{c'} \langle \boldsymbol{m}_c, h(\boldsymbol{x}) \rangle + b_{c'} \to \arg\min_{c'} \|h(\boldsymbol{x}) - \boldsymbol{\mu}_{c'}\|_2$$

where $\tilde{\boldsymbol{\mu}}_c = (\boldsymbol{\mu}_c - \boldsymbol{\mu}_G)/\|\boldsymbol{\mu}_c - \boldsymbol{\mu}_G\|_2$, $\boldsymbol{M} = [\boldsymbol{\mu}_c - \boldsymbol{\mu}_G, c = 1, ..., C] \in \mathbb{R}^{p \times C}$, $\boldsymbol{W}$ contains the classifier weights and $\delta_{c,c'}$ is the Kronecker delta symbol (Papyan et al., 2020).

## A.3 ALIGNMENT AND ACCURACY

Recall that Proposition 4.1 states:

**Proposition.** *Consider a network $f$ with a last-layer linear classifier. Assume the model is trained to neural collapse with cross-entropy loss, regularization and evenly distributed classes. Then letting $\boldsymbol{w}_c$ denote the classifier weight vector corresponding to class $c$, we have:*

$$y = \arg\max_c \boldsymbol{w}_c h(\boldsymbol{x}) + b_c \iff y = \arg\max_c \|\boldsymbol{w}_c\| \|h(\boldsymbol{x})\| cos(\boldsymbol{w}_c, h(\boldsymbol{x}))$$

The proof for Proposition 4.1 follows.

*Proof.* Under neural collapse with cross-entropy loss, regularization and evenly distributed classes, the bias $\boldsymbol{b}$ equals the zero vector (Hong & Ling, 2024). Then by combining

$$y = \arg\max_c \boldsymbol{w}_c h(\boldsymbol{x}) + b_c = \arg\max_c \boldsymbol{w}_c h(\boldsymbol{x}) \quad \text{and} \quad \cos(\boldsymbol{w}_c, h(\boldsymbol{x})) = \frac{\boldsymbol{w}_c \cdot h(\boldsymbol{x})}{\|\boldsymbol{w}_c\| \|h(\boldsymbol{x})\|}$$

we can conclude that

$$y = \arg\max_c \boldsymbol{w}_c h(\boldsymbol{x}) + b_c \iff y = \arg\max_c \|\boldsymbol{w}_c\| \|h(\boldsymbol{x})\| \cos(\boldsymbol{w}_c, h(\boldsymbol{x}))$$

$\square$

## B REPRODUCTION DETAILS

In this section we present details of our experiments to aid with reproduction. Additionally, please find the code used for our experiments here: https://github.com/awesomealex1/NEO. A large part of our code is based on the repository used by FOA (Niu et al., 2024). The implementation for Surgeon is based on the original paper repository (Ma et al., 2025).

### B.1 TTA METHODS

In our experiments we compare with T3A (Iwasawa & Matsuo, 2021), SAR (Niu et al., 2023), LAME (Boudiaf et al., 2022), TENT (Wang et al., 2021), CoTTA (Wang et al., 2022), FOA (Niu et al., 2024) and Surgeon (Ma et al., 2025). We use the default hyperparameters specified in the papers, unless the default hyperparameters cause catastrophic forgetting (accuracy goes to zero), in which case we modify the method to use hyperparameters that do not cause catastrophic forgetting (as most papers do not have results for all models or datasets that we use).

For Surgeon we set the learning rate of Adam to $10^{-5}$. For TENT we use a learning rate 0.00025 for all datasets, as we keep a batch size of 64 throughout most experiments. The batch size is only reduced for on-device resource consumption (due to memory limitations) and sample/class size experiments (only applicable to NEO).

### B.2 DATASETS

We evaluate on ImageNet-C (50 samples $\times$ 1000 classes $\times$ 15 corruption types) (Hendrycks & Dietterich, 2019), CIFAR-10-C (1000 samples $\times$ 10 classes $\times$ 15 corruption types) (Hendrycks & Dietterich, 2019), ImageNet-Rendition (30,000 samples, 200 classes) (Hendrycks et al., 2021) and ImageNet-Sketch (50 samples $\times$ 1000 classes) (Wang et al., 2019).

CIFAR-10-C is available here: https://zenodo.org/records/2535967. ImageNet-C is available here: https://zenodo.org/records/2235448. ImageNet-Rendition is available here: https://people.eecs.berkeley.edu/hendrycks/imagenet-r.tar. ImageNet-Sketch is available here: https://drive.google.com/file/d/1Mj0i5HBthqH1p_yeXzsg22gZduvgoNeA/view.

For CIFAR-10-C we only use the 15 basic corruption types and not the 4 additional types.

### B.3 MODELS

In our experiments we use three different sizes of Vision Transformer (ViT) (Dosovitskiy et al., 2021): ViT-S, ViT-Base and ViT-L, which have 22, 86, and 307 million parameters and embedding dimensions of 384, 768 and 1024 respectively. We use versions finetuned on ImageNet (Deng et al., 2009) or CIFAR-10 (Krizhevsky & Hinton, 2009).

For models used on ImageNet we obtained model weights from timm (Wightman, 2019). We used 'vit_small_patch16_224', 'vit_base_patch16_224' and 'vit_large_patch16_224'. For models finetuned on CIFAR-10, we used publicly available weights from huggingface: 'MF21377197/vit-small-patch16-224-finetuned-Cifar10', 'nateraw/vit-base-patch16-224-cifar10' and 'tzhao3/vit-L-CIFAR10'. They use the same ViT architecture as our ImageNet ViTs, but are finetuned on CIFAR-10.

### B.4 METRICS

We evaluate accuracy in two ways, depending on the type of experiments.

The first way is that we use the accuracy achieved on samples used during the adaptation process. This means that the model starts out unadapted (resulting in potentially low accuracy) and adapt over time (increasing accuracy).

The second way is that we use the accuracy achieved on data not used for adaptation, after finishing the adaptation process. This means we split the dataset into an adaptation set and a validation set. We then calculate accuracy on the validation set, only after adaptation is finished.

We evaluate model calibration using ECE (Pakdaman Naeini et al., 2015), which quantifies how well a model's assigned probabilities align with the actual correctness. ECE is computed by grouping predictions into bins (in our case 15) based on the confidence of the prediction. The difference between observed accuracy and average confidence is calculated and then a weighted average is taken. A low ECE signifies good calibration while a high one implies bad calibration that is overconfident on wrong predictions or under-confident on correct predictions.

Table 3: Accuracy (%) with standard deviations across different corruption types and adaptation methods with ResNet-50 on ImageNet-C. The highest accuracy per corruption type is in bold, and the second-highest is underlined. 512 samples used for adaptation and level 5 corruptions.

| Corruption | No Adapt | SAR | TENT | CoTTA | NEO | NEO-BN |
|---|---|---|---|---|---|---|
| *Noise* | | | | | | |
| Gaussian | 20.96 (0.64) | 14.00 (1.09) | 14.00 (1.09) | 13.80 (1.48) | **23.05 (0.32)** | 13.87 (1.75) |
| Shot | 21.94 (1.12) | 16.34 (1.48) | 16.67 (1.20) | 16.54 (1.33) | **24.35 (1.43)** | 16.73 (1.37) |
| Impulse | 21.94 (0.09) | 13.87 (0.70) | 13.87 (0.16) | 13.80 (0.09) | **23.89 (1.29)** | 13.80 (0.40) |
| *Blur* | | | | | | |
| Defocus | 14.71 (1.34) | 11.65 (0.96) | 12.11 (0.57) | 11.59 (0.33) | **16.93 (1.15)** | 11.85 (1.52) |
| Glass | 8.33 (1.06) | **13.61 (0.51)** | 13.35 (0.40) | 13.15 (0.37) | 9.24 (1.33) | 13.48 (0.70) |
| Motion | 16.93 (0.66) | 23.50 (1.04) | **24.15 (1.45)** | 23.76 (1.90) | 19.47 (1.06) | 23.11 (1.36) |
| Zoom | 19.40 (0.40) | 35.42 (0.56) | **35.61 (0.37)** | 35.48 (0.24) | 21.03 (1.18) | 34.96 (0.32) |
| *Weather* | | | | | | |
| Snow | 23.76 (1.36) | 40.69 (2.23) | **40.82 (2.19)** | 40.43 (2.38) | 26.76 (1.00) | 39.45 (2.51) |
| Frost | 33.01 (1.31) | 41.28 (2.31) | 41.41 (2.32) | **41.47 (2.02)** | 35.61 (0.51) | 40.82 (0.97) |
| Fog | 29.62 (0.24) | 56.90 (0.93) | **57.36 (1.30)** | 57.03 (0.89) | 33.01 (0.89) | 56.12 (1.78) |
| Brightness | 64.65 (1.12) | 66.67 (0.75) | 66.73 (0.66) | **67.06 (0.72)** | 64.71 (1.33) | 65.49 (0.72) |
| *Digital* | | | | | | |
| Contrast | 9.24 (1.13) | **24.87 (0.88)** | 24.67 (0.49) | **24.87 (0.64)** | 10.61 (1.22) | 24.67 (1.15) |
| Elastic | 12.24 (0.80) | 42.25 (1.85) | 42.25 (1.71) | **42.38 (1.52)** | 15.30 (1.06) | 41.41 (1.39) |
| Pixelate | 12.50 (0.48) | 38.02 (1.06) | **38.61 (1.43)** | 38.54 (0.79) | 14.00 (0.09) | 37.17 (1.36) |
| JPEG | 45.90 (0.32) | 40.69 (2.58) | 40.49 (2.44) | 40.49 (2.30) | **46.94 (0.40)** | 39.84 (2.53) |
| **ImageNet-C** | 23.68 | 31.98 | **32.14** | 32.03 | 25.66 | 31.52 |

## B.5 RESOURCE EFFICIENCY

The following component versions were used for experiments on resource usage:

- The Raspberry PI 4B (8GB RAM) used the following software versions: Debian 12 ("bookworm", kernel: 6.6.51+rpt-rpi-v8), Python 3.11.2, torch 2.8.0, torchvision 0.23.0.

- NVIDIA Jetson Orin Nano (8GB) used the following software versions: Ubuntu 20.04.6 LTS (kernel: 5.10.192-tegra), Python 3.8.10, CUDA 11.4, torch 2.1.0a0+41361538.nv23.6, torchvision 0.16.0.

- AMD Instinct MI300X (192GB VRAM) and INTEL(R) XEON(R) PLATINUM 8568Y+ used the following software versions: Ubuntu 24.04.1 LTS, Python 3.12.3, rocm-libs version 6.4.1.60401-83 24.04, torch 2.6.0+rocm6.4.2.git76481f7c, torchvision 0.21.0+rocm6.4.2.git4040d51f.

## C ADDITIONAL RESULTS

### C.1 RESULTS ON CNNS

In Table 3 can see that on ResNet-50, NEO still is able to deliver considerable improvements over using no adaptation. We compare vanilla NEO, as well as NEO-BN, which updates batch normalization parameters, as is common in many other TTA methods (but not applicable to transformers). What is notable, is that in case where using no adaptation is the second most effective, NEO is the most effective method. All other methods, deliver a reduction in accuracy, suggesting that NEO is suitable for adaptation in settings where no other method is able to improve.

### C.2 ABLATION STUDIES

In Table 4a we can see that NEO is batch size independent, and there is no significant difference in final accuracy when changing the batch size. This confirms our theoretical analysis of NEO. When

Table 4: Ablation Studies

| Batch Size | Accuracy (%) |
|---|---|
| 1 | 56.91 |
| 4 | 57.04 |
| 8 | 56.99 |
| 16 | 57.02 |
| 32 | 56.97 |
| 64 | 56.98 |

| Alpha | Accuracy (%) |
|---|---|
| 0.00 | 53.27 |
| 0.01 | 53.40 |
| 0.01 | 53.59 |
| 0.05 | 54.39 |
| 0.10 | 55.14 |
| 0.50 | 56.64 |

(a) Average accuracy across all corruptions for varying batch sizes. ViT-Base on ImageNet-C, using all level 5 corruptions, when adapting with `NEO`. Adapting on 512 samples and evaluating on remaining ones.

(b) Average accuracy across all corruptions for levels of $\alpha$. ViT-Base on ImageNet-C, using all level 5 corruptions, when adapting with `NEO`-Continuous. Adapting on 512 samples and evaluating on remaining ones.

Table 5: Neural Collapse (NC) statistics across different ViT architectures. Measured on ImageNet. Metrics used from (Wu & Papyan, 2024)

| Model | NC1 | NC2 | NC3 | NC4 |
|---|---|---|---|---|
| ViT-Small | 0.6070 | 1.2207 | 1.1969 | 0.8518 |
| ViT-Base | 0.5110 | 1.0092 | 0.9501 | 0.9238 |
| ViT-Large | 0.4640 | 1.2548 | 1.2069 | 0.9067 |

using `NEO`-Continuous, we can see in Table 4b, adapting on 512 samples per corruption, higher $\alpha$ outperforms the lower ones. This is because of the effectiveness of `NEO` in low sample settings, not needing many samples to get the full benefits of adaptation. A lower adaptation rate will only slowly update the model and thus provide less overall accuracy.

## C.3 NEURAL COLLAPSE STATISTICS

In Table 5 we can see the NC statistics for NC1 (Variability Collapse), NC2 (Convergence to simplex ETF), NC3 (Convergence to self-duality) and NC4 (Simplification to nearest class-center). We can see a clear decrease in NC1 as model size increases. NC2 and NC3 follow a similar trend to each other, where ViT-Base performs best. Similarly, NC4 is highest for ViT-Base, closely followed by ViT-Large. Overall, we can see that ViT-Base shows the highest neural collapse.

## C.4 RESOURCE CONSUMPTION ON RASPBERRY PI

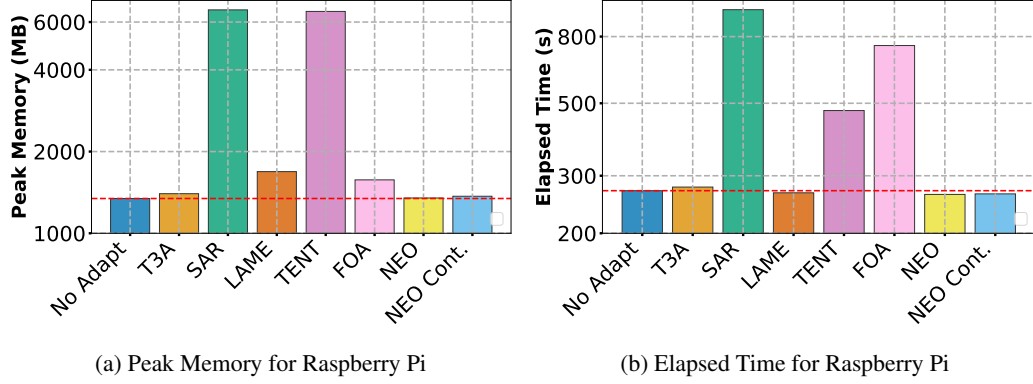

(a) Peak Memory for Raspberry Pi

(b) Elapsed Time for Raspberry Pi

Figure 8: Peak memory and elapsed time for adapting on Vit-Base on ImageNet-C. Raspberry Pi 128 samples - Gaussian Noise 5 - Batch Size = 8.

`NEO` is the most efficient TTA method for both memory usage and inference time. Due to the large memory requirements of CoTTA and Surgeon we could not show results for them on Raspberry Pi.

## C.5 IMAGENET-C BREAKDOWN BY CORRUPTION TYPE

Table 6: Accuracy (%) with 95% confidence intervals across different corruption types and adaptation methods with ViT-Small on ImageNet-C. Accuracy is calculated on the 512 samples used to adapt. The highest accuracy per corruption type is in bold, and the second-highest is underlined.

| Corruption | No Adapt | T3A | SAR | LAME | TENT | CoTTA | FOA | Surgeon | NEO |
|---|---|---|---|---|---|---|---|---|---|
| *Noise* | | | | | | | | | |
| Gaussian | 33.4 (0.6) | 33.0 (1.4) | 33.8 (1.4) | 32.6 (1.3) | 34.7 (1.5) | 33.0 (1.4) | 33.7 (0.8) | 35.3 (2.2) | **36.0 (0.7)** |
| Shot | 32.1 (0.5) | 32.1 (1.0) | **34.7 (0.9)** | 31.8 (0.8) | 34.0 (0.9) | 31.7 (0.7) | 33.4 (0.7) | 34.4 (1.7) | 34.7 (0.6) |
| Impulse | 33.0 (0.5) | 32.7 (0.9) | **35.5 (1.2)** | 32.2 (0.8) | 34.5 (1.0) | 32.8 (1.3) | 34.0 (0.5) | 33.6 (2.6) | 35.5 (0.5) |
| *Blur* | | | | | | | | | |
| Defocus | 30.9 (0.5) | 31.3 (1.2) | 32.0 (1.3) | 30.7 (1.2) | 32.1 (1.5) | 31.9 (0.8) | **37.1 (0.6)** | 32.0 (1.4) | 35.9 (0.5) |
| Glass | 22.9 (0.6) | 23.5 (0.7) | 24.9 (0.7) | 22.7 (0.7) | 24.6 (0.8) | 23.7 (0.8) | 25.1 (0.5) | 24.3 (1.0) | **26.0 (0.6)** |
| Motion | 41.1 (0.6) | 40.7 (1.2) | 41.6 (1.2) | 40.2 (1.1) | 41.9 (1.2) | 40.4 (1.1) | 44.2 (0.5) | 40.8 (1.9) | **44.4 (0.6)** |
| Zoom | 32.5 (0.6) | 32.3 (1.2) | 33.2 (1.2) | 31.7 (1.2) | 33.5 (1.1) | 31.9 (1.0) | 35.2 (0.5) | 32.9 (2.2) | **36.4 (0.6)** |
| *Weather* | | | | | | | | | |
| Snow | 43.6 (0.6) | 44.4 (0.9) | 44.9 (1.0) | 42.1 (0.8) | 44.9 (0.9) | 44.2 (0.9) | 47.2 (0.6) | 43.7 (3.4) | **48.5 (0.5)** |
| Frost | 43.3 (0.7) | 44.2 (1.4) | 44.9 (1.4) | 43.4 (1.4) | 45.0 (1.5) | 43.8 (1.3) | 46.4 (0.6) | 44.6 (1.6) | **47.6 (0.8)** |
| Fog | 46.3 (0.6) | 46.5 (0.9) | 46.3 (1.1) | 45.3 (0.8) | 46.0 (1.0) | 47.0 (1.0) | 49.7 (0.6) | 46.3 (1.6) | **51.9 (0.5)** |
| Brightness | 70.4 (0.6) | 70.8 (0.8) | 71.3 (0.9) | 70.4 (0.9) | 71.4 (0.9) | 70.3 (1.3) | 71.3 (0.7) | 71.5 (2.5) | **71.9 (0.5)** |
| *Digital* | | | | | | | | | |
| Contrast | 16.0 (0.5) | 16.0 (0.8) | 17.9 (2.2) | 15.7 (0.9) | 18.7 (0.9) | 16.1 (0.9) | **21.5 (0.5)** | 15.9 (1.8) | 19.7 (0.4) |
| Elastic | 36.9 (0.6) | 36.9 (1.2) | 36.4 (1.8) | 35.5 (1.2) | 37.6 (1.2) | 37.0 (0.9) | 42.4 (0.6) | 38.3 (3.1) | **43.8 (0.5)** |
| Pixelate | 55.4 (0.6) | 55.6 (1.0) | 56.5 (1.0) | 55.2 (1.1) | 56.9 (1.0) | 56.1 (1.2) | 56.3 (0.7) | 54.8 (2.8) | **57.7 (0.6)** |
| JPEG | 55.2 (0.5) | 55.4 (0.9) | 56.3 (1.0) | 55.0 (0.8) | 56.2 (1.0) | 55.2 (1.1) | **57.9 (0.7)** | 54.3 (2.3) | 57.8 (0.5) |
| **ImageNet-C** | 39.6 (0.6) | 39.7 (1.1) | 40.7 (1.3) | 39.0 (1.0) | 40.8 (1.1) | 39.7 (1.1) | 42.4 (0.6) | 40.2 (2.2) | **43.2 (0.6)** |

Table 7: Accuracy (%) with 95% confidence intervals across different corruption types and adaptation methods with ViT-Large on ImageNet-C. Accuracy is calculated on the 512 samples used to adapt. The highest accuracy per corruption type is in bold, and the second-highest is underlined.

| Corruption | No Adapt | T3A | SAR | LAME | TENT | CoTTA | FOA | Surgeon | NEO |
|---|---|---|---|---|---|---|---|---|---|
| *Noise* | | | | | | | | | |
| Gaussian | 63.1 (0.7) | 63.4 (1.6) | 63.4 (1.5) | 63.0 (1.6) | 63.9 (1.6) | 63.4 (1.5) | 63.1 (0.6) | **65.6 (1.5)** | 64.5 (0.7) |
| Shot | 61.6 (0.6) | 61.1 (1.1) | 61.1 (1.1) | 60.7 (1.0) | 61.9 (1.0) | 61.6 (1.3) | 62.2 (0.7) | **64.6 (2.9)** | 62.9 (0.5) |
| Impulse | 63.7 (0.6) | 64.3 (1.4) | 64.4 (1.3) | 64.0 (1.3) | 64.6 (1.4) | 64.4 (1.1) | 64.0 (0.6) | 64.1 (1.6) | **65.0 (0.6)** |
| *Blur* | | | | | | | | | |
| Defocus | 52.7 (0.7) | 53.1 (1.2) | 53.2 (1.2) | 52.6 (1.2) | 53.9 (1.2) | 52.8 (0.9) | **56.8 (0.5)** | 53.5 (2.1) | 56.2 (0.7) |
| Glass | 44.8 (0.6) | 44.8 (1.0) | 44.9 (1.1) | 44.3 (1.1) | 45.7 (0.9) | 45.2 (1.1) | 46.0 (0.7) | 45.5 (2.3) | **46.4 (0.5)** |
| Motion | 60.5 (0.6) | 60.7 (1.4) | 60.7 (1.3) | 60.3 (1.3) | 61.2 (1.4) | 61.0 (1.3) | 61.8 (0.6) | 60.5 (3.1) | **62.4 (0.6)** |
| Zoom | 55.0 (0.7) | 55.0 (1.7) | 55.2 (1.7) | 54.6 (1.7) | 55.8 (1.7) | 55.5 (1.6) | 56.8 (0.6) | 54.9 (1.5) | **57.0 (0.7)** |
| *Weather* | | | | | | | | | |
| Snow | 66.3 (0.8) | 65.8 (2.0) | 65.9 (2.0) | 65.1 (2.0) | 66.2 (1.9) | 66.2 (1.5) | 66.6 (0.6) | 66.1 (3.2) | **68.1 (0.7)** |
| Frost | 62.3 (0.6) | 62.3 (1.1) | 62.4 (1.1) | 61.7 (1.1) | 62.7 (1.1) | 62.9 (1.1) | 63.9 (0.6) | 62.2 (2.8) | **64.4 (0.7)** |
| Fog | 62.6 (0.5) | 61.9 (1.4) | 62.4 (1.2) | 61.4 (1.2) | 62.2 (1.0) | 62.2 (1.5) | 64.7 (0.5) | 62.1 (2.2) | **67.2 (0.6)** |
| Brightness | 80.2 (0.5) | 80.5 (1.1) | 80.3 (0.9) | 80.1 (1.2) | **80.6 (1.0)** | 80.1 (1.0) | 80.1 (0.6) | 79.7 (2.6) | 80.6 (0.5) |
| *Digital* | | | | | | | | | |
| Contrast | 39.7 (0.5) | 39.2 (1.2) | 42.3 (1.6) | 39.2 (1.2) | 40.4 (1.3) | 39.0 (1.2) | **46.0 (0.6)** | 41.1 (1.5) | 42.9 (0.6) |
| Elastic | 56.0 (0.6) | 55.7 (1.0) | 55.4 (1.2) | 54.7 (1.0) | 56.1 (1.1) | 55.5 (0.9) | 59.0 (0.5) | 54.7 (3.1) | **59.3 (0.6)** |
| Pixelate | 74.9 (0.6) | 75.6 (1.3) | 74.8 (1.8) | 75.4 (1.3) | **76.2 (1.2)** | 74.7 (1.2) | 75.0 (0.4) | 74.4 (2.0) | 75.9 (0.5) |
| JPEG | 72.7 (0.5) | 72.7 (1.0) | 73.0 (1.1) | 72.4 (1.1) | 72.9 (1.1) | 72.2 (1.2) | 73.7 (0.6) | 71.1 (3.0) | **74.1 (0.5)** |
| **ImageNet-C** | 61.1 (0.6) | 61.1 (1.3) | 61.3 (1.4) | 60.6 (1.3) | 61.6 (1.3) | 61.1 (1.2) | 62.6 (0.6) | 61.3 (2.5) | **63.1 (0.6)** |

## C.6 ACCURACY RESULTS ON FULL DATASETS

All results are averaged over seeds 1234, 2020, 9999. Not all TTA methods are available for all experiments.

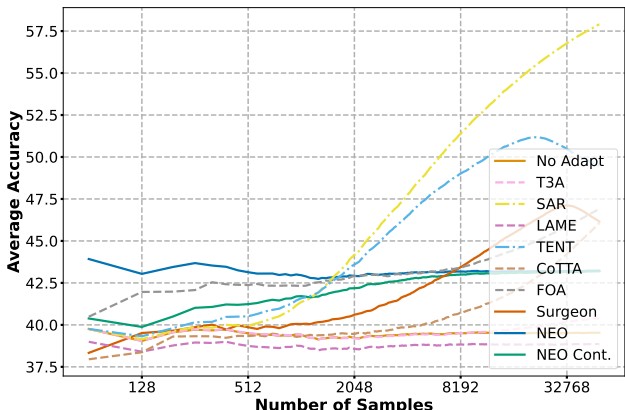

Figure 9: ViT-S - ImageNet-C

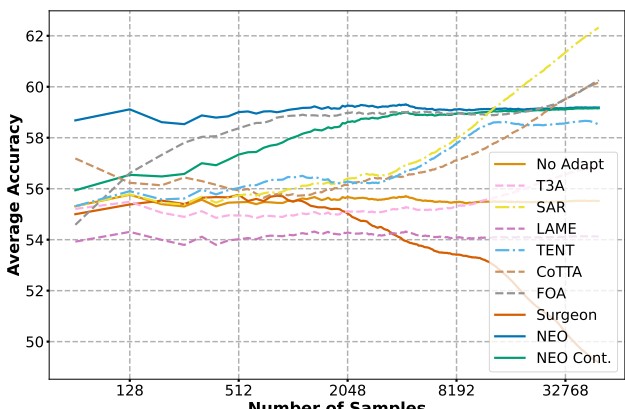

Figure 10: ViT-B - ImageNet-C

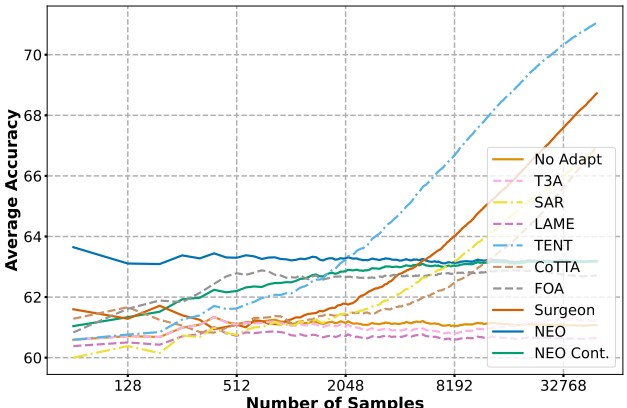

Figure 11: ViT-L - ImageNet-C

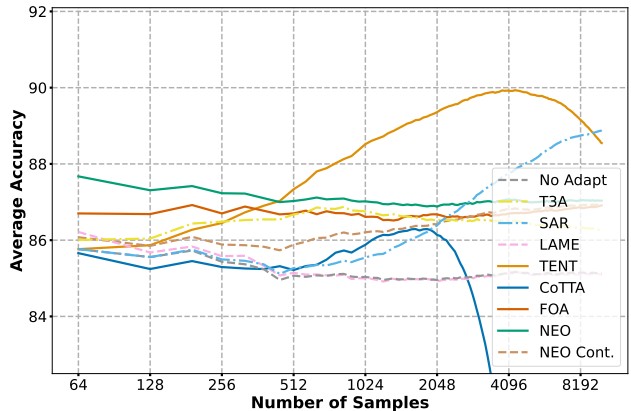

Figure 12: ViT-S - CIFAR-10-C

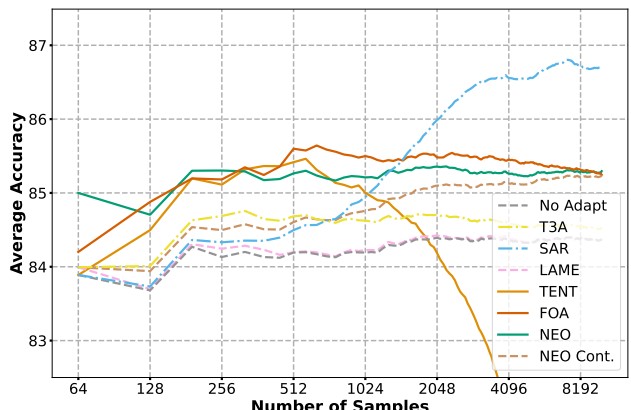

Figure 13: ViT-B - CIFAR-10-C

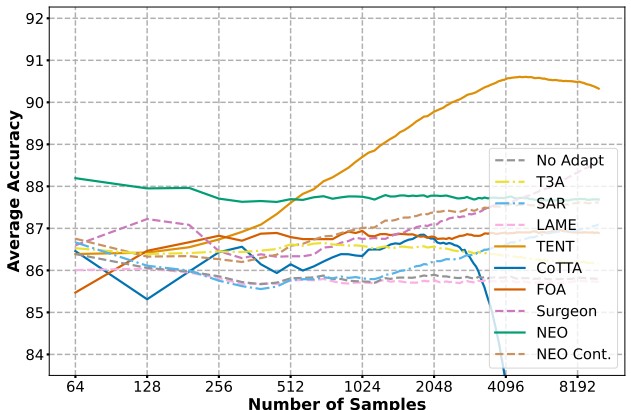

Figure 14: ViT-L - CIFAR-10-C

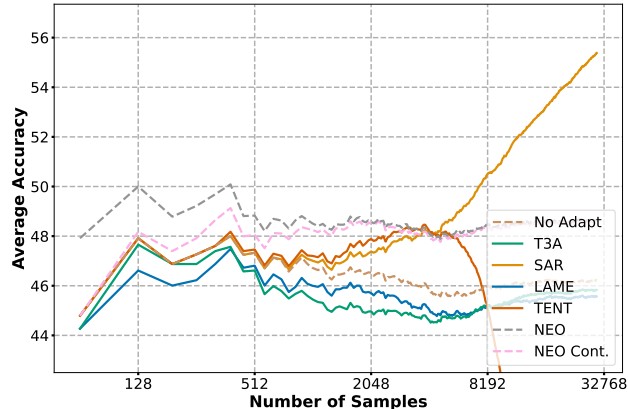

Figure 15: ViT-S - ImageNet-R

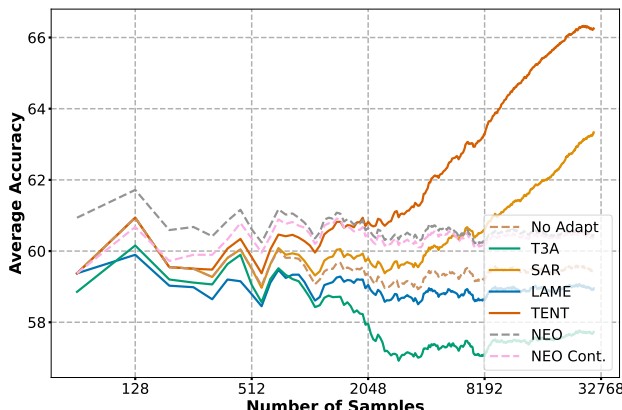

Figure 16: ViT-B - ImageNet-R

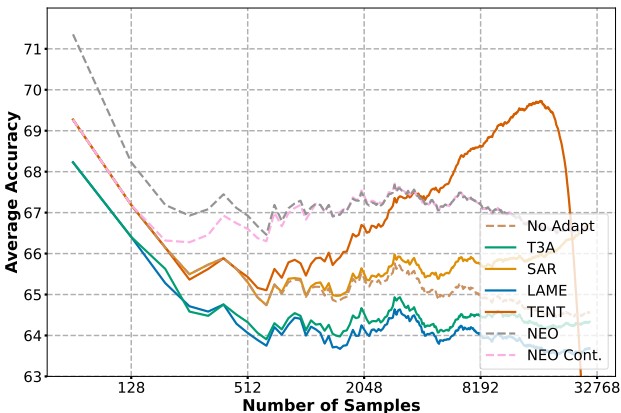

Figure 17: ViT-L - ImageNet-R

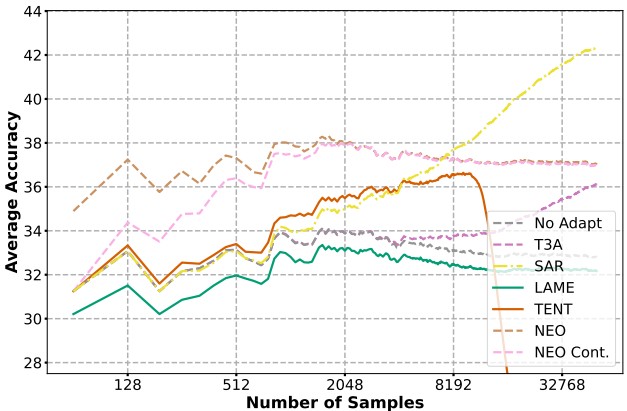

Figure 18: ViT-S - ImageNet-S

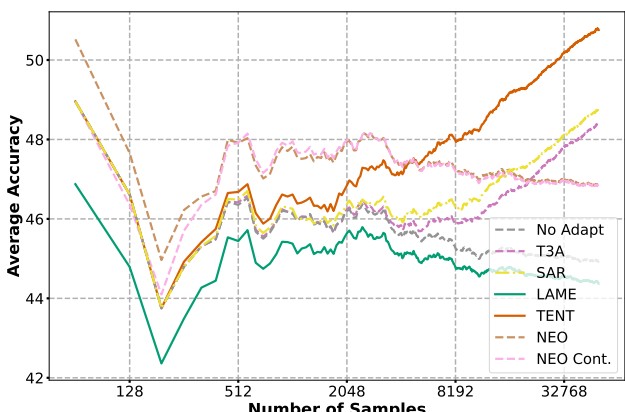

Figure 19: ViT-B - ImageNet-S

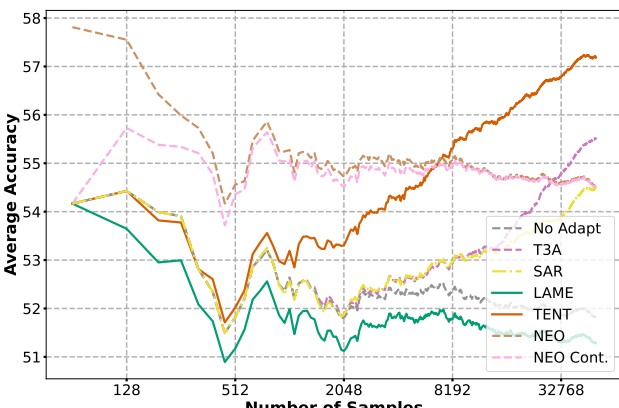

Figure 20: ViT-L - ImageNet-S

## C.7 ECE RESULTS ON FULL DATASETS

Calculated over seeds 1234, 2020 and 9999.

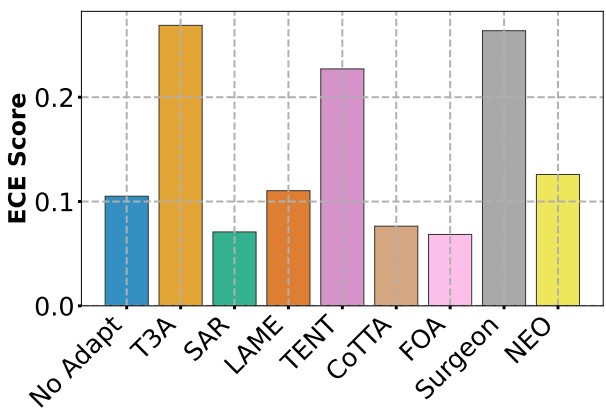

Figure 21: ViT-B - ImageNet-C

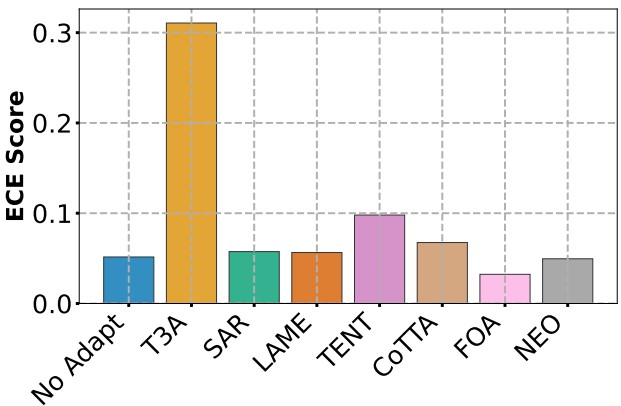

Figure 22: ViT-L - ImageNet-C

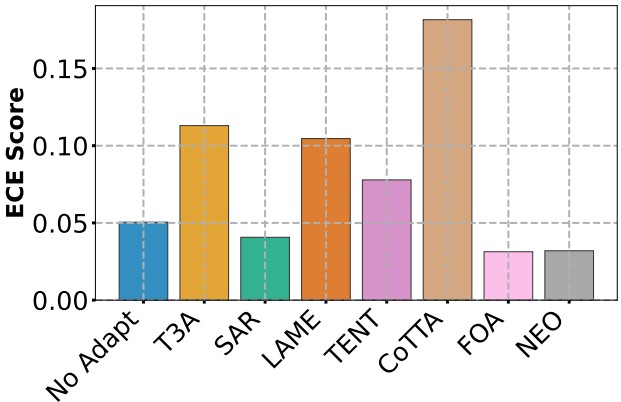

Figure 23: ViT-S - CIFAR-10-C

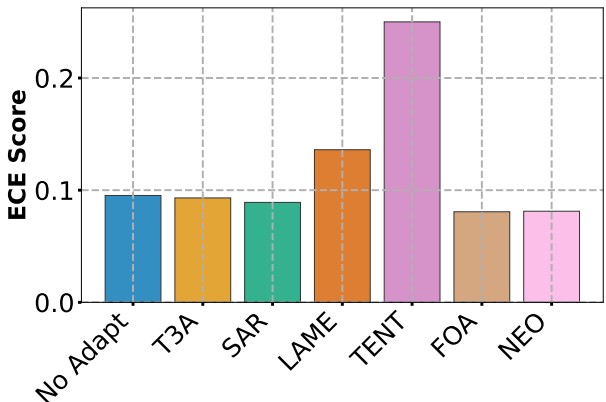

Figure 24: ViT-B - CIFAR-10-C

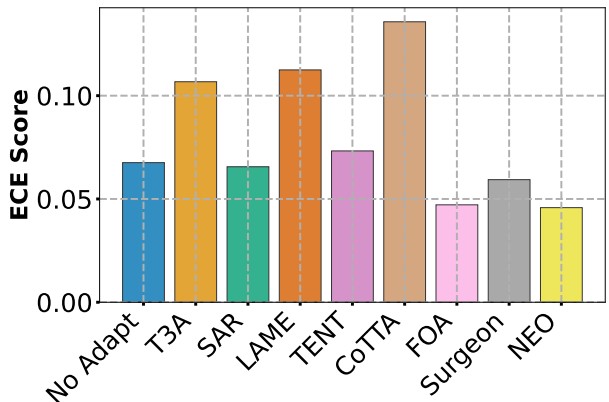

Figure 25: ViT-L - CIFAR-10-C

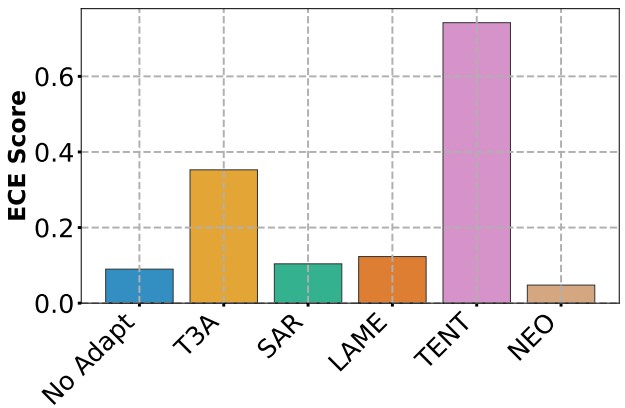

Figure 26: ViT-S - ImageNet-R

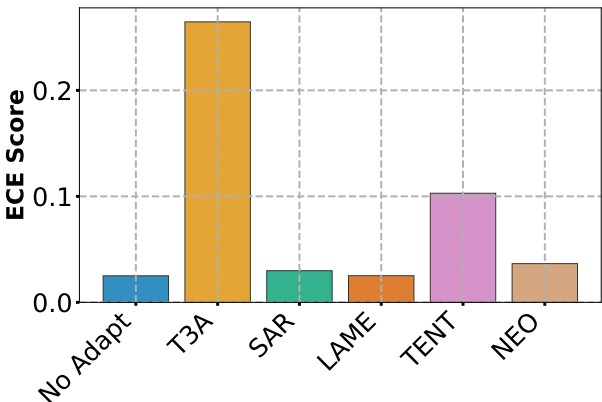

Figure 27: ViT-B - ImageNet-R

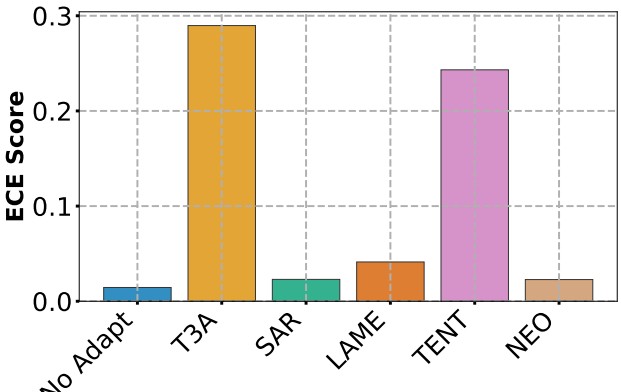

Figure 28: ViT-L - ImageNet-R

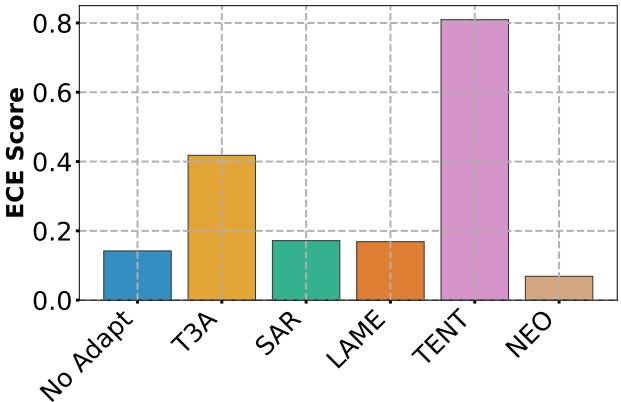

Figure 29: ViT-S - ImageNet-S

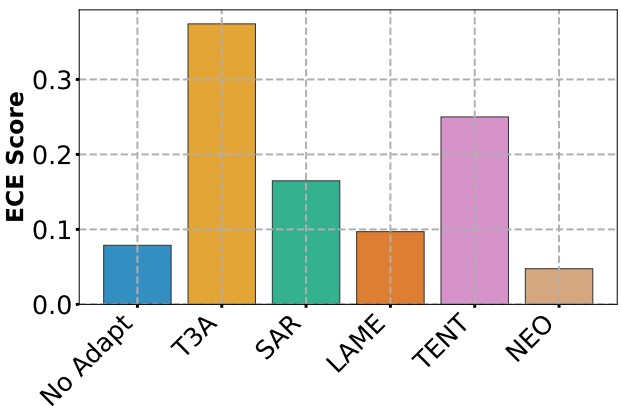

Figure 30: ViT-B - ImageNet-S

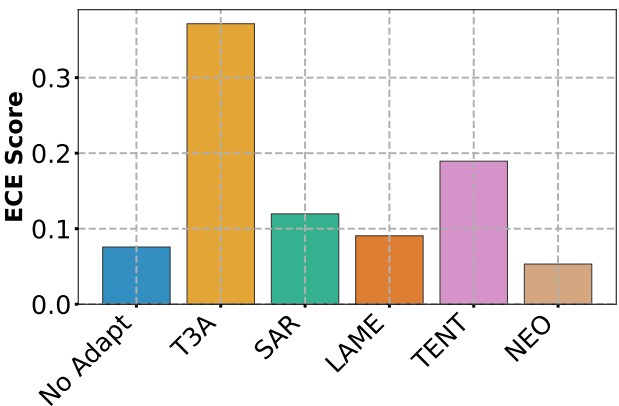

Figure 31: ViT-L - ImageNet-S

## C.8 CONTINUAL ADAPTATION ON IMAGENET-C 512 SAMPLES OVER CORRUPTION INDEX

These figures show adaptation over time (starting adaptation at index 0 and ending at 15). Corruptions are randomly ordered over different repetitions, resulting results that do not depend on a specific sequence of corruptions.

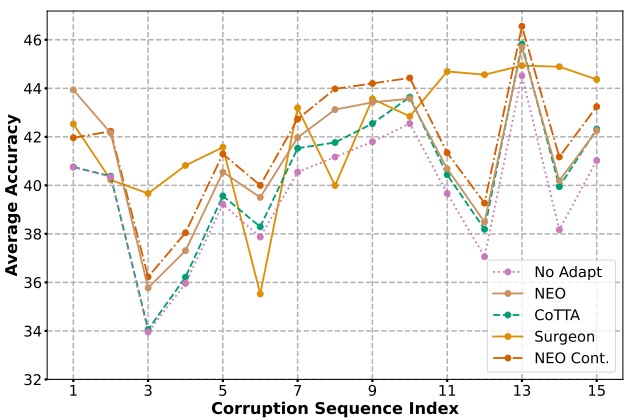

Figure 32: ViT-S - ImageNet-C

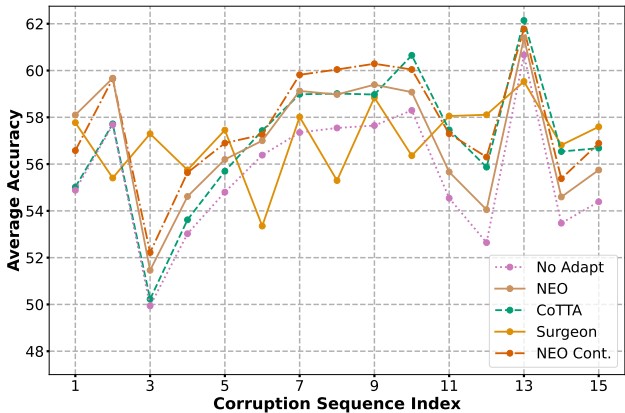

Figure 33: ViT-B - ImageNet-C

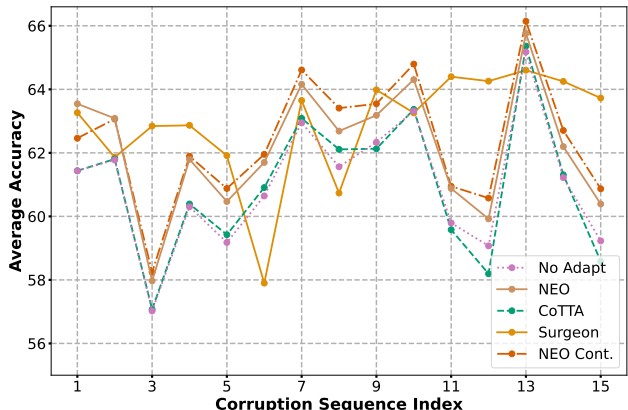

Figure 34: ViT-L - ImageNet-C

# D    DISCLOSURE OF AI USAGE

LLMs were used to help search for relevant works, writing parts of the code (e.g., plots, bash scripts) and proof-reading.

