# OpenReview forum: "NEO — No-Optimization Test-Time Adaptation through Latent Re-Centering"
_ICLR.cc/2026/Conference — ICLR 2026 Poster_

### Official Review · Reviewer_4bB3 · 2025-10-30

**Soundness:** 2
**Presentation:** 2
**Contribution:** 2
**Rating:** 2
**Confidence:** 4

**Summary:**

The paper proposes NEO, a simple test-time adaptation method that performs feature re-centering by subtracting the mean of test embeddings. The method is motivated by the Neural Collapse theory and aims to handle distribution shifts without optimization or access to source data. Experiments are conducted mainly on ImageNet-C and related benchmarks.

**Strengths:**

1. The method is simple and easy to implement.
2. The paper provides a clear and intuitive geometric explanation based on Neural Collapse.
3. Experimental results show that the approach achieves comparable accuracy to some optimization-based TTA methods while requiring less computation.

**Weaknesses:**

1. Lack of methodological novelty. The core idea of this paper’s method is highly similar to the Back-to-Shift mechanism in FOA[1], particularly under the special case where the source feature mean equals zero. Similar effects have already been explored in that setting. The authors are encouraged to further clarify the distinctions between NEO and existing work, and to elaborate on the method’s genuine theoretical or practical contributions.
2. The effectiveness of NEO strongly depends on the assumption that the source-domain embedding mean is zero. If this assumption does not hold (e.g., when models use different normalization mechanisms such as BatchNorm or GroupNorm), the performance of the method may degrade significantly. To verify the generality of this assumption, it is recommended to include additional experiments using models with BatchNorm, GroupNorm, and other normalization layers, and to compare NEO’s robustness across these settings.
3. Insufficient validation of batch-size independence. Despite the claim that the method is batch-size agnostic, the experiments only apply test-time adaptation to a limited number of samples (e.g., a single small batch), and the adapted model is reused for all remaining evaluations—raising questions about true batch-size independence. To substantiate this claim, the authors should perform full test-time adaptation on all test samples using different batch sizes (e.g., 1, 8, 64, 512), where the entire adaptation process is repeated independently for each batch size, and then compare the resulting performance to assess true batch-size independence.
4. Lack of evaluation under label shift scenarios. When class prior distributions differ between the source and target domains, simply subtracting the current batch mean may introduce bias and fail to properly correct distribution shifts. It is recommended to include experiments under label shift settings, as used in studies such as SAR[2], rather than evaluating adaptation only on samples from selected classes.
5. The experiments that adapt using only one batch or one class are compared solely with the “no adaptation” baseline, without including other TTA methods. This limited comparison is insufficient to demonstrate whether NEO still outperforms existing methods under few-sample or class-restricted adaptation scenarios.
6. Insufficient comparison in continual adaptation experiments. In the continual adaptation section, the paper compares NEO with only a few existing methods. It is recommended to include common anti-forgetting TTA methods such as EATA[A], ViDA[B] and KFF[C] to more comprehensively demonstrate NEO’s performance in long-term adaptation scenarios.
7. The method is evaluated using only 512 adaptation samples, which may favor NEO. Optimization-based TTA methods typically employ conservative strategies (e.g., small learning rates) to prevent overfitting. When adaptation data is limited, a slightly higher learning rate may substantially improve their performance. Therefore, this setup may underestimate the performance of other methods.
8. Incomplete disclosure of hyperparameter settings. For some baseline methods—especially those using the same model architecture but with undisclosed configurations—the paper does not provide complete hyperparameter details. To ensure reproducibility and fair comparison, the appendix should include the full hyperparameter settings for each method and model.

[1] Test-Time Model Adaptation with Only Forward Passes. ICML 2024.

[2] Towards stable test-time adaptation in dynamic wild world. ICLR 2023.

[A] Efficient test-time model adaptation without forgetting. ICML 2022.

[B] ViDA: Homeostatic Visual Domain Adapter for Continual Test Time Adaptation. ICLR 2024.

[C] Class-aware Domain Knowledge Fusion and Fission for Continual Test-Time Adaptation. NIPS 2025.

**Questions:**

1. Could the authors leverage the neural collapse assumption to explain why existing entropy minimization–based test-time adaptation (TTA) methods are prone to feature or classifier collapse during adaptation?
2. Could the authors further discuss, from a theoretical perspective, whether the neural collapse assumption still holds—and whether NEO remains effective—in realistic scenarios such as those with class imbalance?

---

> ### Author Response · Authors · 2025-11-21
> **Reply to Reviewer 4bB3 Part 1**
>
> We thank the reviewer for their thoughtful comments.
>
> **Lack of methodological novelty:** Although FOA contains a back-to-source component, NEO differs in the following ways: i) FOA is not source-free, meaning that the back-to-source component relies on source-data that often is not available. This makes NEO applicable in a new setting that FOA is not able to be used in. This is more useful in the current trend of pre-training encoders with multiple source datasets. ii) FOA uses only a few samples of source-data to estimate the source-center. This makes the estimate not as robust as the one of NEO, which is why NEO outperforms FOA. Additionally, the non-continual version of NEO is hyperparameter-free and extremely easy to deploy, making it valuable for edge-device settings for example. We also connect the inner workings of NEO to the theory of neural collapse.
>
> **NEO performance on models with batch norm of group norm:** We ran tests on ResNet-50, where NEO achieved a reduction in mean error on ImageNet-C (15 corruptions at level 5) from 83.23% to 71.74%, when adapting on 512 samples (batch size 64).
>
> | Method | Source | TENT | SAR | CoTTA | LAME | BN Adapt | Surgeon | FOA | NEO |
> | :--- | :--- | :--- | :--- | :--- | :--- | :--- | :--- | :--- | :--- |
> | Mean Error | 83.23% | 80.66% | 70.81% | 70.92% | 94.88% | 70.91% | 69.78% | 84.45% | 71.74% |
>
> **Influence of batch size:** Below are the results for NEO with different batch sizes on Gaussian Noise Level 5 from ImageNet-C with ViT-B on 50k samples, showing no significant impact of batch size:
>
> | Batch Size | 4 | 8 | 16 | 32 | 64 |
> | :--- | :--- | :--- | :--- | :--- | :--- |
> | Accuracy | 56.29 | 56.31 | 56.30 | 56.30 | 56.32 |
>
> In theory, batch size has a negligible effect on NEO because we take an average of all previously seen data to estimate the embedding mean. There is a small advantage at the beginning of adaptation because of a “look-ahead” effect when using a larger batch-size, as the initial embedding mean will be more accurate at the beginning. Over time, this effect is negligible though. In the continual version of NEO the $\alpha$ parameter should be used to counter the effect of batch size.
>
> **Performance degradation with label shift:** We show in our experiments that NEO is able to adapt using samples from just one class (Figure 5 (b)), and improve performance on the remaining 999 classes contained in ImageNet. Further experiments on class imbalance we will leave to future work.
>
> **Comparison with other methods on one-batch or class experiments:** We acknowledge that we didn’t show the performance of other TTA methods in experiments where we only adapt to one batch or a limited number of classes. We do this because we noticed no considerable degradation of NEO’s performance in these scenarios. Since the TTA methods we compare to do not claim improved performance in these scenarios, we do not see reason to believe that they would improve on these more restricted scenarios. We will add results for other methods in the appendix in the final version.
>
> **Comparison with more Continual Adaptation methods:** We experimented on ResNet-50, comparing NEO-Continual and EATA adapting on ImageNet-C in a continual setting. EATA indeed outperforms NEO on 50k samples per corruption (52.05% vs 69.97% mean error), but in the setting with 512 samples, NEO nearly matches EATA (70.43% vs 71.74% mean error), indicating that continual NEO is strong in settings with limited data availability. We will add these results to the final version of the paper.

---

> ### Author Response · Authors · 2025-11-21
> **Reply to Reviewer 4bB3 Part 2**
>
> **Using 512 samples may benefit NEO over other TTA methods:** While NEO performs especially well compared to other TTA methods when using 512 samples, further hyperparameter fine-tuning may improve the performance of other methods. We believe that this shows the importance of making methods that are i) strong in settings where limited data is available for adaptation and ii) have no hyperparameters to have effective adaptation. It is a well-known problem in TTA that many existing methods are unable to adapt to a small number of samples [1]. We believe that NEO provides especially great value in this gap in TTA.
>
> **Incomplete hyperparameter disclosure:** We will outline all complete hyperparameters in addition to those already shown in the Appendix in the final version. The code for NEO, including all hyperparameters, is publicly available: https://anonymous.4open.science/r/NEO-6874/README.md
>
> **Leverage neural collapse to explain why entropy minimization methods are prone to feature or classifier collapse:** We believe that this is a very interesting avenue to pursue in future work, but unfortunately is outside of the scope of this paper.
>
> **Does the neural collapse assumption still hold in scenarios such as class imbalance?** Neural collapse is solely defined by the source data. We show in our experiments (Figure 5 (b)) that NEO can adapt to samples from just a few classes, and improve performance on the remaining classes of the 1000 contained in ImageNet. We intend to leave further experiments on class imbalance for future work.
>
> [1] Michal Danilowski, Soumyajit Chatterjee, and Abhirup Ghosh. Botta: Benchmarking on-device test time adaptation. arXiv preprint arXiv:2504.10149, 2025.

---

### Official Review · Reviewer_rtzZ · 2025-10-31

**Soundness:** 2
**Presentation:** 3
**Contribution:** 2
**Rating:** 4
**Confidence:** 4

**Summary:**

The paper introduces NEO, a hyperparameter-free and optimization-free Test-Time Adaptation (TTA) method that operates by re-centering latent embeddings at the origin. It leverages theoretical insights from neural collapse to justify that centering corrupted test embeddings improves alignment with the clean feature space. NEO requires only forward passes, negligible compute, and no access to source data. Experiments across four datasets (ImageNet-C/R/S, CIFAR-10-C) and three ViT architectures show consistent improvements over prior TTA methods under its own evaluation setting.

**Strengths:**

The studied forward-only test-time adaptation problem is practical and interesting.

NEO’s one-line implementation (replacing nn.Linear with a custom layer) is elegant and simple to use. It adds no backprop, parameters, or optimization loops.

Evaluations on low-power devices (Jetson Orin) highlight NEO’s edge-readiness—rarely studied in TTA works.

**Weaknesses:**

The reported results of the baselines appear questionable. While I understand that the authors may have used different evaluation settings, this explanation is not convincing, as the reported performance is significantly lower than in the original papers. Since this work does not aim to introduce a new evaluation protocol, it should follow the established settings used in prior works such as FOA to ensure fair comparison. As a result, I remain doubtful about the reported results, which undermines confidence in the claimed effectiveness of the proposed method.


The theoretical analysis relies on idealized assumptions—such as balanced classes, cross-entropy training, and zero-centered clean features—which may not fully hold in real-world scenarios. It also remains unclear how well these assumptions apply to models whose features are not zero-centered.

**Questions:**

I am also somewhat confused about the Peak Memory comparison on the Jetson device, as the relative differences between methods seem inconsistent with results typically observed on standard GPUs (e.g., A100).

Additionally, the influence of batch size has not been thoroughly analyzed, particularly for the continual version of NEO.

It would be helpful to clarify whether the method can maintain strong performance under non-i.i.d. settings (e.g., imbalanced label distributions) and whether it also generalizes effectively to CNN-based architectures.

---

> ### Author Response · Authors · 2025-11-21
> **Reply to Reviewer rtzZ**
>
> We thank the reviewer for their thoughtful comments.
>
> **Evaluation settings for baselines:** We followed the same evaluation settings as the baseline papers, except that we used a smaller-sized adaptation data aligned with our primary focus. Additionally, for FOA, we used a population size of k = 2 to ensure that the resource consumption is low and comparable to NEO. Similar resource alignment is impossible for any other methods. In all our experiments, we use the hyperparameters recommended by the authors in their original papers or in code repositories. We list all settings and parameters in the Appendix and in the Anonymized Git repository.
>
> **Memory consumption on A100 vs Jetson:** We believe that the memory consumption will follow a similar trend in server GPUs compared to Jetson, especially because Jetson relies on GPUs, unlike our Raspberry Pi experiments. We will provide memory consumption metrics for A100 in the final version of the paper.
>
> **Influence of batch size:** Below are the results for NEO with different batch sizes on Gaussian Noise Level 5 from ImageNet-C with ViT-B on 50k samples, showing no significant impact of batch size:
>
> | Batch Size | 4 | 8 | 16 | 32 | 64 |
> | :--- | :--- | :--- | :--- | :--- | :--- |
> | Accuracy | 56.29% | 56.31% | 56.30% | 56.30% | 56.32% |
>
> In theory, batch size has a negligible effect on NEO because we take an average of all previously seen data to estimate the embedding mean. There is a small advantage at the beginning of adaptation because of a “look-ahead” effect when using a larger batch-size, as the initial embedding mean will be more accurate at the beginning. Over time, this effect is negligible though. In the continual version of NEO the $\alpha$ parameter should be used to counter the effect of batch size.
>
> **Strong performance on non-i.i.d setting:** We show in our experiments (Figure 5 (b)) that NEO can adapt to samples from just a few classes, and improve performance on the remaining classes of the 1000 contained in ImageNet. We intend to leave further experiments on class imbalance for future work.
>
> **NEO on ResNet:** We ran tests on ResNet-50, where NEO achieved a reduction in mean error on ImageNet-C (15 corruptions at level 5) from 83.23% to 71.74%, when adapting on 512 samples (batch size 64).
>
> | Method | Source | TENT | SAR | CoTTA | LAME | BN Adapt | Surgeon | FOA | NEO |
> | :--- | :--- | :--- | :--- | :--- | :--- | :--- | :--- | :--- | :--- |
> | Mean Error | 83.23% | 80.66% | 70.81% | 70.92% | 94.88% | 70.91% | 69.78% | 84.45% | 71.74% |

---

### Official Review · Reviewer_HPPs · 2025-10-31

**Soundness:** 3
**Presentation:** 3
**Contribution:** 3
**Rating:** 6
**Confidence:** 3

**Summary:**

This paper proposes NEO, a hyperparameter-free test-time adaptation (TTA) method that addresses distribution shifts by re-centering corrupted embeddings at the origin. The core contribution is leveraging neural collapse theory to justify a simple yet effective approach: computing the global centroid of target embeddings and subtracting it to align features with source data. NEO requires no backpropagation, adds minimal computational overhead (storing only a single vector), and demonstrates competitive or superior accuracy compared to seven baseline TTA methods on ImageNet-C, CIFAR-10-C, ImageNet-R, and ImageNet-S using Vision Transformer architectures. The method achieves 59.2% accuracy on ImageNet-C with ViT-Base (vs. 55.6% without adaptation), operates efficiently on edge devices, and maintains robustness even when adapting with as few as one sample or one class.

**Strengths:**

**Originality:** Novel connection between neural collapse theory and TTA. Propositions 4.1-4.2 provide theoretical justification showing $\mu_G = 0_d$ under neural collapse, making $\Delta_G = \tilde{\mu}_G$. The empirical finding that shifts affect <50 of 768 dimensions for 80% of samples motivates the global alignment strategy effectively.

**Quality:** Comprehensive evaluation across three ViT architectures, four datasets, seven baselines. Strong robustness demonstrated: single-sample adaptation, continual learning, edge deployment (Raspberry Pi, Jetson). Figure 3b validates theory with 0.49 cosine similarity to source. Cross-corruption analysis (Figure 6) provides practical deployment insights.

**Clarity:** Well-structured progression from empirical observations â†’ theoretical justification â†’ algorithm. Figure 1 illustrates single-line implementation elegantly. Algorithm 1 is concise and clear.

**Significance:** Addresses critical TTA limitations (cost, memory, hyperparameters) with strong practical value. Achieving highest accuracy (59.2% vs. 58.4% FOA) while being 63% faster demonstrates real-world applicability.

**Weaknesses:**

**Unverified Theoretical Assumptions:** Theory relies on neural collapse, but no empirical validation that ViT models exhibit NC1-NC4 properties (within-class collapse, equiangular tight frame, $\mu_G \approx 0_d$). Proposition 4.2's unconstrained features and balanced class assumptions are strong yet unverified. Need: (a) measure NC metrics on evaluation models, or (b) ablations showing robustness when assumptions fail. Class imbalance in real deployments contradicts balanced class requirement.

**Architecture Specificity:** Claims architecture-agnostic but only evaluates ViTs. CNNs (different inductive biases) may not exhibit same latent properties. Given TENT/SAR target CNNs, this gap is critical. Need: test on at least one CNN (ResNet-50) or explicitly restrict claims to Transformers.

**Missing Failure Analysis:** NEO loses to Surgeon on 3/15 corruptions (Gaussian/Shot/Impulse) and FOA on brightness, with no explanation. Theory assumes global shifts, but what about class-specific components? Figure 3b shows class-wise $\Delta_c$ gains marginal improvement (0.64 vs. 0.51), suggesting untapped potential.

**Weak FOA Comparison:** FOA is most similar (backprop-free, 2 forward passes, efficiency-focused). Paper doesn't explain conceptual differences or why NEO wins (59.2% vs. 58.4%). Is the gain from no-source-data or better adaptation? Need ablation: give NEO source statistics like FOA to isolate re-centering contribution.

**Continual Hyperparameter Contradiction:** NEO-Continual uses EMA parameter $\alpha$, contradicting "hyperparameter-free" claim. No guidance on setting $\alpha$, no sensitivity analysis despite continual adaptation being a key contribution. Need: sensitivity across $\alpha \in [0.01, 0.9]$ or explicit limitation acknowledgment.

**ViT-Base Anomaly Unexplained:** Figure 4a shows ViT-Base gains less than ViT-S/L consistently, contradicting theory that dimensionality shouldn't matter. This pattern deserves investigation.

**Questions:**

1. Can you measure NC1-NC4 metrics on your ViT models? Specifically: $\Sigma_W \approx 0$, equiangular class means, self-duality, $\mu_G \approx 0_d$? How robust is NEO when these fail?

2. Have you tested NEO on CNNs (ResNet-50, EfficientNet)? Would you expect similar effectiveness if neural collapse differs?

3. How does performance degrade with class imbalance? At what ratio does $\mu_G = 0_d$ break down?

5. Is NEO's advantage over FOA from (a) no source data or (b) better adaptation? Can you ablate with source statistics?

6. How to set $\alpha$ in NEO-Continual? Show sensitivity analysis across $\alpha \in [0.01, 0.9]$.

7. Why does ViT-Base underperform ViT-S/L (Figure 4a)?

9. Can pre-computed corruption taxonomy enable zero-shot adaptation (Figure 6)?

10. Quantify exact FLOPs overhead for NEO vs. vanilla inference.

---

> ### Author Response · Authors · 2025-11-21
> **Reply to Reviewer HPPs Part 1**
>
> We thank the reviewer for their thoughtful comments.
>
> **Unverified Theoretical Assumptions and Measurement of Neural Collapse Metrics:** We would like to thank the reviewer for pointing this out. We measured neural collapse metrics for all three types of ViT on both ImageNet and CIFAR-10. Overall, we find that the models trained on CIFAR-10 exhibit clearer neural collapse than on ImageNet, but there is a large variance even within different models trained on the same dataset. We notice the strongest collapse on ViT-B on CIFAR-10. ViT-L, on the other hand, shows weaker collapse. The larger accuracy increase (Figure 4 (a)) on ViT-L, despite the network being less collapsed, indicates that NEO does not rely on the network exhibiting unreasonably strong collapse. We use the collapse metrics for NC1 proposed by [1], for NC2 and NC3 proposed by [2] and for NC4 proposed by [3], all implemented in https://github.com/rhubarbwu/neural-collapse.
>
> | Model | NC1 Variability (↓) | NC2 ETF Error (↓) | NC3 Self Duality (↓) | NC4 Agreement (↑) |
> | :--- | :--- | :--- | :--- | :--- |
> | **ViT-S IN** | 0.38 | 1.16 | 1.20 | 0.89 |
> | **ViT-B IN** | 0.29 | 0.96 | 0.99 | 0.94 |
> | **ViT-L IN** | 0.29 | 1.21 | 1.20 | 0.94 |
> | **ViT-S C-10** | 0.23 | 0.46 | 0.30 | 0.99 |
> | **ViT-B C-10** | 0.04 | 0.07 | 0.08 | 0.99 |
> | **ViT-L C-10** | 0.11 | 0.25 | 0.28 | 0.99 |
>
> **Architecture Specificity:** We conducted experiments on ResNet-50, yielding positive results that demonstrate our method's generality, with an average improvement of 11.49% on ImageNet-C. We will incorporate these experiments into the main text prior to publication.
>
> | Method | Source | TENT | SAR | CoTTA | LAME | BN Adapt | Surgeon | FOA | NEO |
> | :--- | :--- | :--- | :--- | :--- | :--- | :--- | :--- | :--- | :--- |
> | Mean Error | 83.23% | 80.66% | 70.81% | 70.92% | 94.88% | 70.91% | 69.78% | 84.45% | 71.74% |
>
> **Missing Failure Analysis:** NEO underperforms compared to Surgeon in three synthetic corruption types (Table 2). As NEO and Surgeon do not belong to the same family of methods,  they are difficult to compare directly in terms of their behaviour towards any particular corruption types. However, Surgeon is more resource-intensive than any baseline we tested (Figure 7), and more compute allows for better accuracy gains.
>
> NEO falls within the confidence bounds of FOA on brightness, but the mean accuracy of NEO actually is better than that of FOA on the brightness corruption (78.3% vs 78.2%).
>
> **Possible exploration for class-specific alignment:** While we agree that there is an improvement in alignment when using class-specific shifts, during testing, we have found that it is not effective to use the class-specific shift for several reasons. Firstly, identifying the class centers would rely on pseudo-labelling, which is often inaccurate. Secondly, the robustness of NEO would deteriorate because we would only have a few samples per class center to estimate the shift. T3A is a common method that uses class prototypes in a similar way, but NEO outperforms it, showing that the additional robustness of NEO translates to performance (even when a lot of data is available, as shown in the Appendix).
>
> [1] Galanti, T., Gyorgy, A., & Hutter, M. (2021). On the Role of Neural Collapse in Transfer Learning. ArXiv, abs/2112.15121.
>
> [2] Kothapalli, V., Rasromani, E., & Awatramani, V. (2022). Neural Collapse: A Review on Modelling Principles and Generalization. Trans. Mach. Learn. Res., 2023.
>
> [3] V. Papyan, X.Y. Han & D.L. Donoho. (2020). Prevalence of neural collapse during the terminal phase of deep learning training, Proc. Natl. Acad. Sci. U.S.A. 117 (40) 24652-24663.

---

> ### Author Response · Authors · 2025-11-21
> **Reply to Reviewer HPPs Part 2**
>
> **Weak FOA Comparison:** We acknowledge that FOA contains a back-to-source component, but NEO differs in the following ways. Firstly, FOA is not source-free, meaning that the back-to-source component relies on source data that often is not available. This makes NEO applicable in a new setting that FOA is not able to be used in. Secondly, FOA uses only a few samples of source data to estimate the source-center. This makes the estimate less robust than that of NEO, which is why NEO outperforms FOA.
>
> **Continual Hyperparameter Contradiction and Guidance on setting $\alpha$:** We will make it clear in the final version that the continual version of NEO (NEO-Continual) does include one hyperparameter to balance the EMA. Intuitively if one knows how often the domain shifts, more frequent shifts translate to a greater $\alpha$. Essentially, we aim to compute the embedding shift statistics from the part of the history where the domain matches the current batch. $\alpha$ controls the history. We ran NEO-Continual with ViT-Base on ImageNet-C (Level 5), adapting on 512 samples per corruption, with learning rates 0.01, 0.1, 0.25, and 0.5. We get accuracies of 45.70%, 55.31%, 56.77% and 58.28% respectively. In the final version of the paper, we will provide a more detailed study on $\alpha$. We do not attempt to provide a universal $\alpha$ since it is application-specific.
>
> **Why ViT-Base has the smallest gain:** We agree that this potentially is an interesting way to understand the effect of NEO in different architectures. However, we believe that the findings will not enhance the understanding of the core methodology; therefore, we reserve this investigation for the final version of the paper, if accepted.
>
> **NEO on ResNet:** We ran tests on ResNet-50, where NEO achieved a reduction in mean error on ImageNet-C (15 corruptions at level 5) from 83.23% to 71.74%, when adapting on 512 samples (batch size 64).
>
> | Method | Source | TENT | SAR | CoTTA | LAME | BN Adapt | Surgeon | FOA | NEO |
> | :--- | :--- | :--- | :--- | :--- | :--- | :--- | :--- | :--- | :--- |
> | Mean Error | 83.23% | 80.66% | 70.81% | 70.92% | 94.88% | 70.91% | 69.78% | 84.45% | 71.74% |
>
> **Performance degradation with class imbalance:** We show in our experiments that NEO is able to adapt using samples from just one class (Figure 5 (b)), and improve performance on the remaining 999 classes contained in ImageNet. Further experiments on class imbalance we will leave to future work.
>
> **Main advantage over FOA:** NEO has two main advantages over the back-to-source approach from FOA. Firstly, NEO is fully source-free, not needing any samples from the source distribution. This makes NEO applicable in a whole different setting than FOA. Secondly, NEO continuously updates the shifted mean estimate, making it extremely robust. And Finally and most importantly, NEO is optimization-free and thus extremely resource-efficient (as evidenced in Figure 7), unlike FOA, which needs optimization.
>
> **Can NEO be used for zero-shot adaptation:** Precomputed means can be used to adapt, when the model is identical to the one used to calculate the mean. Preliminary experiments have shown that the mean embeddings do not translate between different models.
>
> **Quantify FLOP overhead:** In Figure 7, we show that neither memory consumption or runtime differs to a significant extent compared to no adaptation. Furthermore, NEO does not involve any compute heavy operations, and thus we do not expect FLOP to add any more new insight into the resource consumption.

---

### Meta-Review · Area_Chair_p5qC · 2026-01-11

**Summary:**

Three expert reviewers with relevant publications are split between acceptance (HPPs: 6) and rejection (rtzZ: 4, 4bB3: 2). The strengths include the simplicity and computational efficiency of the method alongside the thorough scope of the benchmarking w.r.t comparisons, while the weaknesses include unverified theoretical assumptions, a narrow focus on one type of architecture (ViTs), potential issues with the soundness of the benchmarking, and its extreme simplicity which could be regarded as not having technical novelty or depth especially in relation to the back-to-source transformation of the existing method FOA. The rebuttal responds to many of the points from the reviewers with clarifications and experiments. As the rebuttal is focused, and specifically addresses reviewer requests with mostly positive resolutions, it is not unlikely that scores would have improved. After a careful examination of the submission, reviews, and the rebuttal responses the meta-reviewer sides with acceptance. While the proposed NEO is indeed extremely simple, this is part of its empirical value, in that it can serve as a reality check on more complex and computationally expensive methods. Nevertheless the detailed and justified feedback from reviewers, especially concerning the further documentation of experiments and the new results, should be incorporated into the work so that it more surely informative and convincing for the community.

**Reviewer Concerns:**

- Exclusively experimenting with ViTs and not convolutional architectures (HPPs, rtzZ): The experiments in the submission only evaluate ViTs. The rebuttal provides positive results with ResNet-50, a common convolutional architecture specifically requested by review, and fully resolves this point.
- Validity of theoretical assumptions (HPPs, rtzZ, 4bB3): Assumptions like standard/clean features of zero mean may not hold and are not measured. The rebuttal provides the specific measurements requested by HPPs but does not discuss with rtzZ which models fit the idealized assumptions or not and how results are impacted when they do not hold. A new result for a different normalization, group normalization, shows that NEO still helps as evidence of flexibility w.r.t. assumptions. This is partially resolved and the included rebuttal results are straightforward to incorporate into the main text.
- Comparability and the Soundness of Evaluations (rtzZ): Comparisons of existing methods like FOA report lower results than the original papers. While there are some identified differences in evaluation setting, this may still indicate faults with the evaluation, which could be addressed by comparing in more existing settings as well. The rebuttal further explains the differences, but does not provide more benchmarking results. This is only partially resolved, and so this submission is borderline because this work is empirical. However, this is not sufficient justification for rejection, because the soundness is not so severely in question.
- Missing experiments for the settings of non-i.i.d. and/or continual adaptation (rtzZ, 4bB3): The non-i.i.d. in the case of missing labels is reported in the paper with improvement over the source model without adaptation. The continual setting is reported in the main text, with positive results, and extended in the rebuttal with mixed results: in sum NEO is competitive with older methods like EATA when the data per domain is limited but then underperforms when more data is available. NEO does still improve over no adaptation.
- Sensitivity to Batch Size (rtzZ, 4bB3): The proposed NEO relies on the batch to estimate the target mean for centering the data. The rebuttal provides a new experiment showing insensitivity across common batch sizes for test-time adaptation and particularly small batch sizes that cause issues for some methods. This is resolved.
- Lack of novelty especially w.r.t. FOA (4bB3, HPPs): The review argues that the centering of the representations is highly similar to the back-to-source transformation. The rebuttal clarifies small but meaningful differences that actually alter the setting and affect the resuilts between the existing FOA and proposed NEO. This is resolved, in that there is a difference from FOA, but only partially resolved, if the issue is the extreme simplicity of NEO.
- Further questions about different settings or characteristics of the data (rtzZ, 4bB3): Reviewers ask about class imbalance, label shift, and more. The rebuttal provides clarifications for each point. Some points are addressed by experiment and others by discussion. One question about a potential explanation for some behaviors of entropy minimizations is fairly declared out of scope although it is nevertheless interesting.These are resolved.

**Reviewer Scores:**

- HPPs: the score of 6 would at least be maintained and more likely would be raised to 7. The rebuttal to this review is clear, addresses requests exactly, and provides new results that resolve the issues raised.
- rtzZ: the score of 4 could increase to 6 in going from marginal rejection to marginal acceptance. The rebuttal addresses most points fully including specific results requested by the review. However, it could do more to address the point about confidence of soundness, because it does not confirm the (re-)evaluation of the submission reproduce the expected results in existing settings (when configured for those settings). For this reason I think the reviewer would increase, but not past marginal acceptance. Nevertheless I think an increase is likely because the reasons for differences in results are clarified, and the full details in the appendix and code are underlined.
- 4bB3: the score of 2 is a strong vote for rejection and in general such scores may not change. In this case, there are multiple dimensions of weakness raised—insufficient novelty, comparisons, and detail for reproduction—and so there is much to address. Nevertheless the rebuttal does address all of these points, and provides new results that defuse several potential issues, so that a higher score is possible. I expect the the score could be raised to 4: this respects that most issues are clarified by explanation or resolved by additional results while acknowledging the first weakness concerning novelty can remain in a reasonablel difference of opinion.

---

### Decision · Program_Chairs · 2026-01-26

Accept (Poster)